# When Combinatorial Thompson Sampling meets Approximation Regret

**Pierre Perrault**
Idemia
pierre.perrault@idemia.com

## Abstract

We study the Combinatorial Thompson Sampling policy (CTS) for combinatorial multi-armed bandit problems (CMAB), within an *approximation regret* setting. Although CTS has attracted a lot of interest, it has a drawback that other usual CMAB policies do not have when considering non-exact oracles: for some oracles, CTS has a poor approximation regret (scaling linearly with the time horizon $T$) [Wang and Chen, 2018]. A study is then necessary to discriminate the oracles on which CTS could learn. This study was started by Kong et al. [2021]: they gave the first approximation regret analysis of CTS for the *greedy oracle*, obtaining an upper bound of order $\mathcal{O}\big(\log(T)/\Delta^2\big)$, where $\Delta$ is some minimal reward gap. In this paper, our objective is to push this study further than the simple case of the greedy oracle. We provide the first $\mathcal{O}(\log(T)/\Delta)$ approximation regret upper bound for CTS, obtained under a specific condition on the approximation oracle, allowing a reduction to the exact oracle analysis. We thus term this condition REDUCE2EXACT, and observe that it is satisfied in many *concrete* examples. Moreover, it can be extended to the *probabilistically triggered arms* setting, thus capturing even more problems, such as *online influence maximization*.

## 1 Introduction

Stochastic multi-armed bandits (MAB) [Robbins, 1952, Berry and Fristedt, 1985, Lai and Robbins, 1985] are decision-making problems in which an *agent* acts sequentially in an uncertain environment. At each round $t \in \mathbb{N}^*$, the agent must select one arm from a fixed set of $n$ arms, denoted by $[n] \triangleq \{1, \ldots, n\}$, using a *policy*, based on the feedback from the previous rounds. Then it gets as feedback an *outcome* $X_{i,t} \in \mathbb{R}$ — a random variable sampled from $\mathbb{P}_{X_i}$, independently from previous rounds — where $i$ is the selected arm and $\mathbb{P}_{X_i}$ is a probability distribution — unknown to the agent — of mean $\mu_i^*$. The goal for the agent is to maximize the *cumulative reward* over a total of $T$ rounds ($T$ is the *time horizon* and may be unknown). The performance metric of a policy is its *regret* $R_T$, which is the expectation of the difference over $T$ rounds between the cumulative reward of the policy that always picked the arm with the highest expected reward and the cumulative reward of the learning policy. MAB models the so called dilemma between exploration and exploitation, i.e., whether to continue exploring arms to obtain more information (and thus strengthen the confidence in the estimates of the distributions $\mathbb{P}_{X_i}$), or to use the information gathered by playing the best arm according to the observations so far.

In this paper, we study stochastic combinatorial multi-armed bandit (CMAB), with *semi-bandit feedback*, a.k.a. stochastic semi-bandit (still abbreviated as CMAB in this paper), an extension of MAB where the agent plays an *action* (also called *super-arm*) $A_t \in \mathcal{A} \subset \mathcal{P}([n])$ at each round $t$, where $\mathcal{A}$ is fixed and called *action space*. The feedback includes the outcomes of all base arms in

36th Conference on Neural Information Processing Systems (NeurIPS 2022).

the played super-arm.[1] The expected reward, given $A_t$, is assumed to be in the form[2] $r(A_t, \boldsymbol{\mu}^*)$, where $\boldsymbol{\mu}^* \in \mathbb{R}^n$ is the unknown vector of expectations (traditionally, the reward is linear and equal to $\mathbf{e}_{A_t}^\top \boldsymbol{\mu}^*$). In recent years, CMAB has attracted a lot of interest (see e.g. Cesa-Bianchi and Lugosi [2012], Gai et al. [2012], Chen et al. [2013, 2016], Kveton et al. [2015], Wang and Chen [2017], Perrault [2020]), particularly due to its wide applications in network routing, online advertising, recommender system, influence marketing, etc.

Many CMAB policies are based on the *Upper Confidence Bound* (UCB) approach, extending the classical UCB policy [Auer et al., 2002] from MAB to CMAB. This type of approach uses an optimistic estimate $\boldsymbol{\mu}_t$ of $\boldsymbol{\mu}^*$ (i.e., for which the reward function is overestimated), lying in a well-chosen confidence region. Then, the action is chosen by plugging $\boldsymbol{\mu}_t$ inside an *oracle* (typically, Oracle($\boldsymbol{\mu}^*$) is a maximizer of the reward function $A \mapsto r(A, \boldsymbol{\mu}^*)$). An example of such policy is *Combinatorial Upper Confidence Bound* (CUCB) [Chen et al., 2013, Kveton et al., 2015], that uses a Cartesian product of the individual confidence intervals of each arm as a confidence region. For mutually independent arms, Combes et al. [2015] provided the UCB-style policy *Efficient Sampling for Combinatorial Bandit* (ESCB), building a tighter axis-aligned ellipsoidal confidence region around the empirical mean, which helps to better restrict the exploration. Degenne and Perchet [2016] provided a policy called OLS-UCB, leveraging a *sub-Gaussianity* assumption on the arms to generalize the ESCB approach. These policies have been further extended to more general settings afterwards [Perrault et al., 2020c,d, 2019a]. Although improving CUCB, all these generalizations are inefficient in terms of computation time.

Another paradigm that has recently gained interest (and which will be our focus in this paper) is to rely on *Thompson Sampling* (TS) instead of UCB, still targeting *frequentist regret*. Although introduced much earlier by Thompson [1933], the theoretical analysis of TS for MAB is quite recent: Kaufmann et al. [2012], Agrawal and Goyal [2012] gave a regret bound matching the UCB policy theoretically. Moreover, TS often performs better than UCB in practice, making TS an attractive policy for further investigations. For CMAB, TS extends to *Combinatorial Thompson Sampling* (CTS). In CTS, the unknown mean $\boldsymbol{\mu}^*$ is associated with a belief (a prior distribution, that could be e.g. a product of Beta or Gaussian distributions) updated to a posterior with the Bayes'rule, each time a feedback is received. In order to choose an action at round $t$, CTS draws a sample $\boldsymbol{\theta}_t$ from the current belief, and plays the action given by Oracle($\boldsymbol{\theta}_t$). CTS is an attractive policy because it has similar advantages to the previously mentioned policies working with ellipsoidal confidence regions while being, like CUCB, computationally efficient. Indeed, recently, for mutually independent arms and sub-Gaussian arms respectively, Wang and Chen [2018], Perrault et al. [2020a] proposed tight analyses of CTS.

Unlike UCB-based policies, the analysis of CTS is valid only when Oracle is exact, i.e., when

$$\text{Oracle}(\boldsymbol{\mu}) \in \arg\max_{A \in \mathcal{A}} r(A, \boldsymbol{\mu}).$$

Although this holds true for many combinatorial problem described by the pair $(r, \mathcal{A})$ (we recall that $r$ is usually linear), there exist some problems where the requirement on Oracle has to be relaxed in order to make it tractable. This is usually done considering an $\alpha$-*approximation* oracle [Chen et al., 2013, 2016, Wen et al., 2016], for $\alpha \in (0, 1)$:

$$r(\text{Oracle}(\boldsymbol{\mu}), \boldsymbol{\mu}) \geq \alpha \max_{A \in \mathcal{A}} r(A, \boldsymbol{\mu}). \tag{1}$$

Under an $\alpha$-approximation oracle, the benchmark cumulative reward is the $\alpha$-fraction of the optimal reward, leading to the notion of *approximation regret* [Kakade et al., 2009, Streeter and Golovin, 2009, Chen et al., 2016].

Aware of the limitation of their CTS analysis (that works only with exact oracles), Wang and Chen [2018] also proved, in their Theorem 2, that this limitation is not a technical artifact. More precisely, they provided a specific CMAB instance with an associated approximation oracle, such that CTS on this instance and with this oracle must have a regret scaling linearly in $T$. Although this negative result is of great interest to the research community, some concerns limit its consideration. Indeed,

---

[1]Note that we will consider the *probabilistically triggered arms* extension in this paper, i.e., where the feedback is on triggered arms. For brevity, we do not present this generalization in the introduction.

[2]Henceforth, we typeset vectors and matrices in bold and indicate components with indices, e.g., $\mathbf{a} = (a_i)_{i \in [n]} \in \mathbb{R}^n$. We also let $\mathbf{e}_i$ be the $i^{th}$ canonical unit vector of $\mathbb{R}^n$, and define the *incidence vector* of any subset $A \subset [n]$ as $\mathbf{e}_A \triangleq \sum_{i \in A} \mathbf{e}_i$. We denote by $\mathbf{a} \odot \mathbf{b} \triangleq (a_i b_i)$ the Hadamard product of two vectors $\mathbf{a}$ and $\mathbf{b}$.

not only the CMAB instance provided by Wang and Chen [2018] is actually a MAB one (meaning that there is an efficient oracle that simply enumerates the arms), so the use of an approximation regret is not justified, but above all, both the oracle and the instance are uncommon and designed for the proof.

Interested in the question of whether the example provided by Wang and Chen [2018] is pathological or generalizable, Kong et al. [2021] recently initiated a study, which revealed that linear approximation regret for CTS seems to be pathological. More precisely, they derived a $\mathcal{O}\big(\log(T)/\Delta^2\big)$ bound for the specific case of a *greedy* oracle[3], where $\Delta$ is some reward gap. This result is obtained by bounding the approximation regret by a *greedy regret*, that simply replaces $\alpha \max_{A \in \mathcal{A}} r(A, \boldsymbol{\mu}^*)$ with $r(\mathrm{Oracle}(\boldsymbol{\mu}^*), \boldsymbol{\mu}^*)$, using equation (1). They also gave a tight lower bound on the greedy regret.

In this paper, we want to explore another class of oracles covering more problems in practice. Our goal is to demonstrate that although there are instrumental examples of problems where CTS has a linear regret, the majority of concrete problems do not follow this regime, and are in fact similar to the exact oracle case.

**Contributions**    With a specific condition on $\mathrm{Oracle}$, we describe a general set of CMAB problems where the approximation regret of CTS has a $\mathcal{O}(\log(T)/\Delta)$ bound, improving by a factor $1/\Delta$ over the bound of Kong et al. [2021]. This does not contradict their lower bound, which is focused on the greedy regret. We call this set of CMAB problems REDUCE2EXACT, because, as we will see, a REDUCE2EXACT problem can be approximated using a reduction to sub-problems that can be solved exactly. Our main result is the approximation regret guaranty for REDUCE2EXACT problems, provided in Theorem 2. The REDUCE2EXACT condition on $\mathrm{Oracle}$ is structural and applies notably to greedy algorithms for submodular maximization. In particular, it allows to deal with problems such as *probabilistic maximum coverage*. We note that REDUCE2EXACT is compatible with the *probabilistically triggered arms* setting, which allows us to capture even more problems, such as *online influence maximization* [Wen et al., 2017, Wang and Chen, 2017]. As we want to focus on concrete problems, we provide several other examples that belongs to REDUCE2EXACT: *Metric k-center*, *Vertex cover*, *Max-Cut* and *Travelling salesman problem*.

**Further related work**    We refer the reader to Wang and Chen [2018] for more related work on TS for combinatorial bandits. Briefly, one can mention Gopalan et al. [2014], that gave a frequentist high-probability regret bounds for TS with a general action space and feedback model — Komiyama et al. [2015], that studied TS for the $m$-sets action space — Wen et al. [2015], that studied TS for contextual CMAB problems, using the Bayesian regret metric (see also Russo and Van Roy [2016]).

**Other known limitations of CTS**    Apart from the limitation related to the approximation regret that interests us in this paper, there are some other existing limitations of CTS highlighted in the literature, which we review here: The CTS policy has an exponential constant term in its regret upper bound [Wang and Chen, 2018, Perrault et al., 2020a], and Wang and Chen [2018] proved in their Theorem 3 that this is unavoidable. A similar behavior have been demonstrated in Zhang and Combes [2021], where it is shown that CTS does not scale polynomially in the ambient dimension $n$ in general. In addition, Zhang and Combes [2021] also proved that CTS is not minimax optimal. Actually, they even proved that in high dimensions, the minimax regret of CTS is almost linear in $T$.

**The strengths of CTS**    Despite the weaknesses mentioned above, CTS remains a widely used policy, mainly because of its empirical performance. Indeed, CTS generally outperforms other policies such as CUCB and ESCB [Wang and Chen, 2018, Perrault et al., 2020a]. Moreover, it is relatively simple to implement, and is computationally efficient (just like CUCB). On the theory side, another advantage is that for an exact oracle, CTS is asymptotically quasi-optimal[4] for many settings where CUCB is not, and where ESCB is computationally inefficient [Perrault et al., 2020a]. It would be desirable that these advantages also apply to the case of approximation regret, thus motivating our investigations.

---

[3]It is worth mentioning that this oracle is one of the most common, so it is logical to focus on it first.

[4]This means that it has a distribution-dependent regret upper bound whose leading term in $T$ has an optimal rate, up to a poly-logarithmic factor in $n$.

## 2 Model and definitions

For more generality, we consider the *probabilistically triggered arms* extension of CMAB [Chen et al., 2016, Wang and Chen, 2017], abbreviated to CMAB-T. In this context, the action $A \in \mathcal{A}$ selected is not necessarily equal to the triggered super-arm $S$. More precisely, the action space $\mathcal{A}$ is no longer necessarily a subset of $\mathcal{P}([n])$ and can be infinite. At round $t$, the agent selects $A_t \in \mathcal{A}$, based on the history of observations $\mathcal{H}_t \triangleq \sigma\left(\mathbf{X}_1 \odot \mathbf{e}_{S_1}, \ldots, \mathbf{X}_{t-1} \odot \mathbf{e}_{S_{t-1}}\right)$ and a possible extra source of randomness (we denote by $\mathcal{F}_t$ the filtration containing $\mathcal{H}_t$ and the extra randomness of round $t$ — in particular, $A_t$ is $\mathcal{F}_t$-measurable). Then, an independent sample $\mathbf{X}_t \sim \mathbb{P}_{\mathbf{X}}$, $\mathbf{X}_t \in \mathbb{R}^n$ is drawn and a random subset $S_t \in \mathcal{S} \subset \mathcal{P}([n])$ of arms are triggered ($\mathcal{S}$ is called *super-arm space* or *subset space*). We assume that $S_t$ is drawn independently from a distribution $D_{\text{trig}}(A_t, \mathbf{X}_t)$ and that the outcome of an arm does not depend on whether it is triggered. In addition, if we don't have $\mathbb{P}_{\mathbf{X}} = \otimes_{i \in [n]} \mathbb{P}_{X_i}$, we assume that $D_{\text{trig}}$ doesn't depend on $\mathbf{X}_t$. For the feedback, the outcome of each triggered arm is observed, i.e., $\mathbf{e}_{S_t} \odot \mathbf{X}_t$ is observed. The expected reward is of the form $r(A_t, \boldsymbol{\mu}^*)$, where $r$ is a function defined on a domain $\mathcal{A} \times \mathcal{M}$, with $\mathcal{M} \subset \mathbb{R}^n$. The objects $\mathcal{A}, D_{\text{trig}}, r$ are known to the agent. We assume that $r(\cdot, \boldsymbol{\mu}^*)$ admits a maximum $r(A^*, \boldsymbol{\mu}^*)$ on $\mathcal{A}$. In the following, we give the definition of the probability that an arm $i \in [n]$ is triggered (and thus that a feedback from $i$ is obtained) by having played a certain action $A \in \mathcal{A}$.

**Definition 1** (Triggering probabilities). *The triggering probabilities are defined for all $i \in [n]$ and $A \in \mathcal{A}$ as $p_i(A) \triangleq \mathbb{P}[i \in S]$, where $S \sim D_{trig}(A, \mathbf{X})$, $\mathbf{X} \sim \mathbb{P}_{\mathbf{X}}$.*

Under an $\alpha$-approximation Oracle, we use the approximation regret to evaluate the performance of a policy $\pi$, defined as follows.

**Definition 2** (Approximation regret). *The $T$-round $\alpha$-approximation regret of a learning policy $\pi$ that selects action $A_t \in \mathcal{A}$ at round $t$ is defined as follows, where the approximation gap is defined as $\Delta_t = \Delta(A_t) \triangleq 0 \vee (\alpha r(A^*, \boldsymbol{\mu}^*) - r(A_t, \boldsymbol{\mu}^*))$, with $A^* \in \arg\max_{A \in \mathcal{A}} r(A, \boldsymbol{\mu}^*)$.*

$$R_{T,\alpha}(\pi) \triangleq \mathbb{E}\left[\sum_{t \in [T]} \Delta_t\right].$$

To approach the problem of minimizing $R_{T,\alpha}$, we consider the following standard assumptions [Wang and Chen, 2017].

**Assumption 1** (Approximation oracle). *The agent has access to an* Oracle *such that for any mean vector $\boldsymbol{\mu} \in \mathcal{M}$,*

$$r(\text{Oracle}(\boldsymbol{\mu}), \boldsymbol{\mu}) \geq \alpha r(A^*, \boldsymbol{\mu}).$$

**Assumption 2** (1-norm triggering probability modulated bounded smoothness). *There exists $\mathbf{B} \in \mathbb{R}_+^n$ such that for all $A \in \mathcal{A}$, for all $\boldsymbol{\mu}, \boldsymbol{\mu}' \in \mathcal{M}$,*

$$|r(A, \boldsymbol{\mu}) - r(A, \boldsymbol{\mu}')| \leq \sum_{i \in [n]} p_i(A) B_i |\mu_i - \mu_i'|.$$

**Assumption 3** (Sub-Gaussianity of the outcome distribution). *$\mathbb{P}_{\mathbf{X}}$ is such that $\forall \boldsymbol{\lambda} \in \mathbb{R}^n$,*

$$\mathbb{E}\left[e^{\boldsymbol{\lambda}^\top (\mathbf{X} - \boldsymbol{\mu}^*)}\right] \leq e^{\|\boldsymbol{\lambda}\|_2^2/8}.$$

*For example, $\mathbb{P}_{\mathbf{X}} = \otimes_{i \in [n]} \mathbb{P}_{X_i}$, and $X_i \overset{a.s.}{\in} [0,1]$ (from Hoeffding's Lemma [Hoeffding, 1963]).*

**Definition 3** (Other definitions). *We define, for $i \in [n]$, the minimal gap of an action containing $i$ as*

$$\Delta_{i,\min} \triangleq \inf_{A \in \mathcal{A}: \, p_i(A) > 0, \, \Delta(A) > 0} \Delta(A).$$

*The minimal and maximal gaps are defined as*

$$\Delta_{\min} \triangleq \min_{i \in [n]} \Delta_{i,\min} \quad and \quad \Delta_{\max} \triangleq \sup_{A \in \mathcal{A}} \Delta(A).$$

*For $A \in \mathcal{A}$, we let $\mathrm{T}(A) \triangleq \{i \in [n] : p_i(A) > 0\}$ be the set of arms that are triggerable by selecting action $A$. We finally define*

$$m \triangleq \sup_{A \in \mathcal{A}} |\mathrm{T}(A)|, \quad m^* \triangleq |\mathrm{T}(A^*)|, \quad and \quad p^* \triangleq \inf_{i \in [n], \, A \in \mathcal{A}: \, p_i(A) > 0} p_i(A).$$

---
**Algorithm 1** CTS-BETA
---
**Initialization**: For each arm $i$, let $\gamma_i = \delta_i = 1$.
**For all** $t \geq 1$:
   Draw $\boldsymbol{\theta}_t \sim \otimes_{i \in [n]} \text{Beta}(\gamma_i, \delta_i)$.
   Play $A_t = \text{Oracle}(\boldsymbol{\theta}_t)$.
   Get the observation $\mathbf{X}_t \odot \mathbf{e}_{S_t}$, and draw $\mathbf{Y}_t \sim \otimes_{i \in S_t} \text{Bernoulli}(X_{i,t})$.
   For all $i \in S_t$ update $\gamma_i \leftarrow \gamma_i + Y_{i,t}$ and $\delta_i \leftarrow \delta_i + 1 - Y_{i,t}$.
---

---
**Algorithm 2** CTS-GAUSSIAN
---
**Input**: $\beta > 1$.
**Initialization**: Play each arm once (if the agent knows that $\boldsymbol{\mu}^* \in [a, b]^n$, this might be skipped)
**For every subsequent round** $t$:
   Draw $\boldsymbol{\theta}_t \sim \otimes_{i \in [n]} \mathcal{N}\left(\overline{\mu}_{i,t-1}, N_{i,t-1}^{-1} \beta/4\right)$ ($\theta_{i,t} \sim \mathcal{U}[a, b]$ if $N_{i,t-1} = 0$).
   Play $A_t = \text{Oracle}(\boldsymbol{\theta}_t)$.
   Get the observations $\mathbf{X}_t \odot \mathbf{e}_{S_t}$ and let $\mathbf{Y}_t = \mathbf{X}_t$.
   Update $\overline{\boldsymbol{\mu}}_{t-1}$ and counters accordingly.
---

## 3   Combinatorial Thompson Sampling and exact oracle analysis

In this section, we present the CTS policy, focusing on two versions, one working with a Beta prior, and the other with a Gaussian prior. Then, we present an associated analysis for the exact oracle case (i.e., with $\alpha = 1$).

### 3.1   Algorithms

Based on the above assumptions, we focus on two versions of CTS. The first version is CTS-BETA [Wang and Chen, 2018] (Algorithm 1), working when we assume furthermore that $\mathbb{P}_{\mathbf{X}} = \otimes_{i \in [n]} \mathbb{P}_{X_i}$ and $\mathbf{X} \overset{a.s.}{\in} [0, 1]^n$ (note that this actually covers the case of bounded $\mathbf{X}$, by adjusting the parameter $\mathbf{B}$). For each arm $i \in [n]$, CTS-BETA maintains a Beta prior distribution with parameters $\gamma_i$ and $\delta_i$ (initialized to 1). At each round $t$, for each arm $i$, the algorithm sample $\theta_{i,t}$ from the corresponding prior, representing the current estimate of $\mu_i^*$. Then the oracle outputs the action $A_t$ to play according to the input vector $\boldsymbol{\theta}_t$. Based on the observation feedback, the algorithm then updates the corresponding Beta distributions. The second version is CTS-GAUSSIAN [Perrault et al., 2020a] (Algorithm 2), that works under the more general Assumption 3. It is essentially the same as CTS-BETA, except that the prior distributions are Gaussian. For both Algorithm 1 and Algorithm 2, and an arm $i \in [n]$, we define the number of time $i$ has been triggered at the beginning of round $t$, called counter of arm $i$, as

$$N_{i,t-1} \triangleq \sum_{t' \in [t-1]} \mathbb{I}\{i \in S_{t'}\}.$$

We also define the *empirical mean* at the beginning of round $t$ as

$$\overline{\mu}_{i,t-1} \triangleq \sum_{t' \in [t-1]} \frac{\mathbb{I}\{i \in S_{t'}\} Y_{i,t'}}{N_{i,t-1}}.$$

### 3.2   Analysis of CTS: the $\alpha = 1$ case

Although this is close to some known results in the current literature [Huyuk and Tekin, 2019, Perrault et al., 2020a], there is no proof for the classical $\mathcal{O}\left(\log(m) \log(T) \sum_{i \in [n]} B_i^2 / \Delta_{i,\min}\right)$ regret bound under the above assumptions in the CMAB-T setting, either for CTS-BETA (Algorithm 1) or CTS-GAUSSIAN (Algorithm 2). We thus provide such a result in Theorem 1 (the proof is postponed to Appendix A). We can notice a difference with the work of Huyuk and Tekin [2019] concerning the Assumption 2, where the triggering probabilities do not appear (and are present in the main term of their final regret bound). The main difference with Perrault et al. [2020a] is that they do not consider probabilistically triggered arms.

**Theorem 1.** *If $\Delta_{\min} > 0$ and $p^* > 0$, the policy $\pi$ described in Algorithm 1 (under Assumptions 1, 2 and $\mathbb{P}_{\mathbf{X}} = \otimes_{i \in [n]} \mathbb{P}_{X_i}$, $\mathbf{X} \overset{a.s.}{\in} [0,1]^n$) or Algorithm 2 (under Assumptions 1, 2 and 3) has a regret of order*

$$R_{T,1}(\pi) = \mathcal{O}\left( \sum_{i \in [n]} \frac{B_i^2 \log(m) \log(T)}{\Delta_{i,\min}} \right).$$

In addition to being a new result in itself, Theorem 1 will be useful for the $\alpha < 1$ case. Concerning the bound, after changing the CMAB setting to CMAB-T, it should be noticed that the $T$-independent additive constant depends on $1/p^*$, the cause being the use of the Lipschitz condition (Assumption 2) without weighting by probabilities. We think that this dependence should be avoidable, but it seems that another technique has to be considered. Finally, we remark that this kind of bound can be usually transformed into a gap-independent $\sqrt{T}$ bound [Chen et al., 2013], however, for CTS, achieving this transformation is impossible since CTS is not minimax optimal (as we mentioned in the paragraph "Other known limitations of CTS").

## 4  The $\alpha < 1$ case for REDUCE2EXACT problems

We will now look at the $\alpha < 1$ case. Our strategy is based on the following observation: many approximation algorithms involve a relaxation, or a reduction to one or more problems that can be solved exactly (in this paper, we use the terminology sub-problems to refer to them). The approximation guarantee for such an approximation algorithm is thus obtained by linking the original problem to those sub-problems. We give a simple abstract example to illustrate this idea. Let's say we want to maximize a function $f$ on some set $F$, using an $\alpha$-approximation algorithm. Assume there exist two other functions $g$ and $h$ defined on sets $G$ and $H$ respectively, such that $G \times H \subset F$ and such that $g$ and $h$ can be maximized exactly on $G$ and $H$ respectively. Finally, assume that for all $(x, y) \in G \times H$,

$$\alpha \max_F f - f(x, y) \leq \max_G g - g(x) + \max_H h - h(y). \tag{2}$$

This means that an $\alpha$-approximation algorithm to maximize $f$ can simply output the feasible solution $(\arg\max_G g, \arg\max_H h) \in F$. This example may seem very basic and artifactual at first glance, but it turns out that many approximation algorithms rely on the same principle, as we will see in subsection 4.2. To see how this can be exploited for approximation regret minimization, we can notice that the LHS of (2) is an approximation gap (as defined in Definition 2, taking the max with 0 and considering that the choice $(x, y) \in F$ is that of a policy at a given round, with $F$ playing the role of the action space) and that the RHS is the sum of two gaps, with $G$ (respectively $H$) playing the role of the action space, but this time without approximation factor. Summing over the rounds, we find that the corresponding approximation regret is bounded by the sum of two "classical" regrets (in this paper, we use the terminology sub-regrets). Thus, the bounds we obtain on these two sub-regrets using CTS with the corresponding exact oracles translate into an bound on the approximation regret.

**Avoiding the "mismatch" phenomenon**  Kong et al. [2021] identified the reason why their regret bound has a $\Delta^2$ in the denominator, while the usual CMAB algorithms only have a $\Delta$. They term it a "mismatch" between the estimated gaps that need to be eliminated by exploration and the actual regret the algorithm needs to pay. We argue that, in fact, this mismatch phenomenon should exist in principle, even for non-approximation regret. It is usually avoided using a smoothness assumption like Assumption 2, linking our estimations (here the arms that generate the feedback) and what is paid. In the above example, although we are in an approximation context, the situation is fundamentally no different. We see that (2) links the paid approximation regret with two exact sub-regrets which are themselves assumed to be related to our estimates through a smoothness-like property. There is thus an indirect link between the approximation regret and the outcomes, which enables one of the $\Delta$ present in the denominator of the exploration term $\log(T)/\Delta^2$ to be cancelled out by the actual regret paid, thus avoiding the mismatch phenomenon.

To summarize, just as the approximation relation is obtained by linking the original problem to sub-problems that can be solved exactly, the idea behind REDUCE2EXACT problems is to link the approximation regret to several sub-regrets, each satisfying the appropriate properties for CTS, namely a smoothness relationship and the availability of an exact oracle. We formalize this in the following assumption.

**Assumption 4** (REDUCE2EXACT). *There exist $\ell \in \mathbb{N}^*$, $\mathbf{c} \in \mathbb{R}_+^\ell$ and $\mathbf{B}_j \in \mathbb{R}_+^n$ for all $j \in [\ell]$ such that the following is true.* Oracle *is of the form* Oracle $=$ Oracle$_2 \circ$ Oracle$_1$, *where* Oracle$_1$ *and* Oracle$_2$ *are described as follows.*

- Oracle$_1$ : *For $\boldsymbol{\mu} \in \mathcal{M}$,* Oracle$_1(\boldsymbol{\mu})$ *must output a sequence $(E_1, \ldots, E_\ell)$ described as follows. For each $j \in [\ell]$, let $\mathcal{E}_j = \mathcal{E}_j(E_1, \ldots, E_{j-1})$ be a sub-action space which may depend on $E_1, \ldots, E_{j-1}$ and let $r_j(\cdot, \boldsymbol{\mu}) : \mathcal{E}_j \to \mathbb{R}$ be a reward sub-function. Then, we require that $E_j \in \arg\max_{E \in \mathcal{E}_j} r_j(E, \boldsymbol{\mu})$.*

- Oracle$_2$ : *For an input $E_1 \in \mathcal{E}_1, \ldots, E_\ell \in \mathcal{E}_\ell$,* Oracle$_2(E_1, \ldots, E_\ell)$ *must output an action in $\mathcal{A}$ such that:*

$$\Delta(\text{Oracle}_2(E_1, \ldots, E_\ell)) \leq \sum_{j \in [\ell]} \left( r_j\left(E_j^*, \boldsymbol{\mu}^*\right) - r_j(E_j, \boldsymbol{\mu}^*) \right) \cdot c_j, \qquad (3)$$

*where for all $j \in [\ell]$, $E_j^* \in \arg\max_{E \in \mathcal{E}_j} r_j(E, \boldsymbol{\mu}^*)$.*

- *Finally, in addition to the above constraints on* Oracle$_1$ *and* Oracle$_2$, *for each $j \in [\ell]$, we require that the reward sub-function $r_j$ satisfies Assumption 2 with the constants $\mathbf{B}_j$ and with the triggering probabilities $p_i(\text{Oracle}_2(E_1, \ldots, E_\ell))$, $i \in [n]$.*

Informally, in the above Assumption 4, for $\boldsymbol{\mu} \in \mathcal{M}$, Oracle$_1(\boldsymbol{\mu})$ exactly solves a finite sequence of recursively defined optimization sub-problems and Oracle$_2$ builds an action in $\mathcal{A}$ for the original approximation problem using the intermediate solutions provided by Oracle$_1$. At first sight, Assumption 4 seems very specific and rather difficult to fulfill. However, we will see in subsection 4.2 that many concrete problems satisfy it.

## 4.1 Analysis

In this subsection, we give in Theorem 2 the main result of this paper. It basically states that under Assumptions 4, the CTS policy have a regret bound comparable to the exact oracle case. The proof is postponed to Appendix B.

**Theorem 2.** *If $\Delta_{\min} > 0$ and $p^* > 0$, the policy $\pi$ described in Algorithm 1 (under Assumption 4 and $\mathbb{P}_{\mathbf{X}} = \otimes_{i \in [n]} \mathbb{P}_{X_i}$, $\mathbf{X} \stackrel{a.s.}{\in} [0,1]^n$) or Algorithm 2 (under Assumptions 3 and 4) has regret of order*

$$R_{T,\alpha}(\pi) = \mathcal{O}\left( \sum_{i \in [n]} \frac{\left( \sum_{j \in [\ell]} B_{ij} c_j \right)^2 \log(m) \log(T)}{\Delta_{i,\min}} \right).$$

The idea of the proof is quite simple once Assumption 4 has been made. We can see that the approximation regret can be decomposed into $\ell$ sub-regrets, according to equation (3). Then, one must focus on the fact that the sub-regrets may dependent on each other and that the right gap (defined with the original reward function) must be obtained in the denominator of the final bound.

## 4.2 Examples of REDUCE2EXACT problems

Here, we present several problems belonging to REDUCE2EXACT. Each time, after a quick introduction of the problem, we translate it into our CMAB-T context, and finally show how it satisfies the REDUCE2EXACT criteria. The Travelling salesman problem (TSP) is treated in Appendix C.

**Submodular maximization (e.g., probabilistic maximum coverage (PMC))** Here, we only expose the PMC example, noting that the same derivation can be applied to a monotone submodular[5] function. PMC is one of the main examples proposed by Kong et al. [2021]. Given a weighted bipartite graph $G = (L, R, E)$, with weights $\boldsymbol{\mu}^* \triangleq (\mu^*_{(u,v)})_{(u,v) \in E}$ (notice there are thus $n = |E|$ arms, recalling that for us an arm is only something that produces an outcome, not something we can choose as an action, explaining why arms are not indexed by vertices here), the goal is to find an

---

[5]$f$ is monotone if for every $A \subset B$, we have $f(A) \leq f(B)$. It is submodular if for every $A, B$ we have that $f(A \cup B) + f(A \cap B) \leq f(A) + f(B)$.

action $A \in \mathcal{A} \triangleq \{A \subset L : |A| = k\}$, $k \in \mathbb{N}^*$, maximizing the expected number of influenced nodes in $R$, where each node $v \in R$ can be independently influenced by $u \in A$ with probability $\mu^*_{(u,v)}$, i.e., maximizing $f(A, \boldsymbol{\mu}^*) \triangleq \sum_{v \in R} \left(1 - \prod_{u \in A: (u,v) \in E}(1 - \mu^*_{(u,v)})\right)$. This problem can be applied to the semi-bandit framework called *the ad placement problem*, where $L$ are the web pages, $R$ are the users and $\mu^*_{(u,v)}$ is the probability that user $v$ clicks on the ad on web page $u$. In this application, the user's click probabilities are unknown and must be learned as the rounds progress. The Greedy oracle can provide an approximate solution with approximation ratio $\alpha = 1 - 1/e$ [Nemhauser et al., 1978]. This setting fits REDUCE2EXACT as follows:

- $\text{Oracle}_1(\boldsymbol{\mu})$ : For $i \in [k]$, let $\mathcal{E}_i \triangleq \{(a_1, \ldots, a_i) : (a_1, \ldots, a_{i-1}) = E_{i-1}, a_i \in L \backslash E_{i-1}\}$ and $r_i((a_1, \ldots, a_i), \boldsymbol{\mu}) \triangleq f(\{a_1, \ldots, a_i\}, \boldsymbol{\mu})$.

- $\text{Oracle}_2(E_1, \ldots, E_k)$ : Let $(a_1, \ldots, a_k) \triangleq E_k$. $\text{Oracle}_2$ returns $A = \{a_1, \ldots, a_k\}$.

Let $A^i = \{a_1, \ldots, a_i\}$ for some $i \in [k]$ and by abuse of notation, let $f = f(\cdot, \boldsymbol{\mu}^*)$. Informally, we see that in the above decomposition of the oracle, at each step, $\text{Oracle}_1$ maximizes $f(A^i)$ with $A^{i-1}$ fixed, i.e., $\text{Oracle}_1$ optimizes only on $a_i$ (we thus recover the greedy algorithm). It is precisely these sub-problems of finding $a_i$ that can be solved exactly. Then, we see that $\text{Oracle}_2$ simply returns the last $A^i$ constructed. We will now prove the relation (3). The following is true using that $f$ is monotone submodular (this is actually the way Nemhauser et al. [1978] proved the approximation guarantee, and is true for any monotone submodular function):

$$f(A^*) - f(A^i) \leq \sum_{a \in A^* \backslash A^i} \left(f(\{a\} \cup A^i) - f(A^{i+1})\right) + k\left(f(A^{i+1}) - f(A^i)\right).$$

Once the above relation is obtained, we can actually continue the original proof from Nemhauser et al. [1978], skipping each step where we would need to use the property of $\text{Oracle}_1$, thus leaving a term in the right-hand side for each time we skipped.

$$f(A^*) - f(A^k) = f(A^*) - f(A^{k-1}) - \left(f(A^k) - f(A^{k-1})\right)$$
$$\leq (f(A^*) - f(A^{k-1}))\left(1 - \frac{1}{k}\right) + \frac{1}{k} \sum_{a \in A^* \backslash A^{k-1}} \left(f(\{a\} \cup A^{k-1}) - f(A^k)\right)$$
$$\ldots \leq f(A^*)\left(1 - \frac{1}{k}\right)^k + \sum_{i=1}^{k} \frac{\left(1 - \frac{1}{k}\right)^{i-1}}{k} \sum_{a \in A^* \backslash A^{k-i}} \left(f(\{a\} \cup A^{k-i}) - f(A^{k-i+1})\right).$$

Finally, since $\left(1 - \frac{1}{k}\right)^k \leq e^{-1}$, we get that $\Delta(\text{Oracle}_2(E_1, \ldots, E_k)) = (1 - e^{-1})f(A^*) - f(A)$ is bounded by

$$\sum_{i=1}^{k} \frac{\left(1 - \frac{1}{k}\right)^{i-1}}{k} \left|A^* \backslash A^{k-i}\right| \left(r_{k-i+1}(E^*_{k-i+1}, \boldsymbol{\mu}^*) - r_{k-i+1}(E_{k-i+1}, \boldsymbol{\mu}^*)\right).$$

Note in passing that we recover the classical approximation if the right-hand-side was equal to 0. It is easy to see that the reward function $r_j$ satisfies Assumption 2 with $\mathbf{B}_j = \mathbf{e}_{[n]}$. We thus finally get our Assumption 4.

**Online influence maximization (OIM)**   As the analysis mentioned above only uses submodularity, it can be extended to the problem of online influence maximization in a social network. A social network is modeled as a directed graph $G = (V, E)$, with nodes $V$ representing users and edges $E$ representing connections. For a node $i \in V$, a subset $A \subset V$, and a vector $\mathbf{x} \in \{0, 1\}^E$, let the predicate $A \xrightarrow{\mathbf{x}} i$ hold if, in the graph defined by $G_{\mathbf{x}} \triangleq (V, \{ij \in E, x_{ij} = 1\})$, there is a forward path from a node in $A$ to the node $i$. If it holds, we say that $i$ is influenced by $A$ under $\mathbf{x}$. The goal is to find an action $A \in \mathcal{A} \triangleq \{A \subset V : |A| = k\}$ maximizing the *influence spread* $\sigma(A, \boldsymbol{\mu}^*) \triangleq \mathbb{E}\left[\left|\left\{i \in V, A \xrightarrow{\mathbf{X}} i\right\}\right|\right]$, where $\mathbf{X} \sim \otimes_{(u,v) \in E} \text{Bernoulli}(\mu^*_{(u,v)})$. This model is called the *independent cascade model* [Kempe et al., 2003, 2015]. A notable property to use the greedy oracle is that $\sigma$ is monotone submodular. As the exact calculation of $\sigma$ is prohibitive, it is estimated by simulating the diffusion process, resulting in an approximation factor $\alpha = 1 - e^{-1} - \varepsilon$ in the

above greedy oracle analysis [Kempe et al., 2015, Feige, 1998, Chen et al., 2010], with $\varepsilon > 0$. In OIM, arms may be probabilistically triggered, and Assumption 2 holds with the constants being all equal to $\max_{u \in V} \left| \left\{ v \in V, \ \{u\} \overset{(\mathbb{I}\{\mu_e^* > 0\})_{e \in E}}{\rightsquigarrow} v \right\} \right|$, which is the largest number of nodes any node can reach [Wang and Chen, 2017]. As previously, we thus get Assumption 4 with $\mathbf{B}_j$ being $\mathbf{e}_{[n]}$ times this constant.

**Metric $k$-center** This example and the following ones are less common for CMAB, but allow to well illustrate REDUCE2EXACT. Given a set of cities, one wants to build $k$ warehouses in different cities and minimize the maximum distance of a city to a warehouse. Formally, given a complete undirected weighted graph $G = (V, E)$ whose distances $d(v_i, v_j)$ satisfy the triangle inequality, the goal is to find an action $A \in \mathcal{A} \triangleq \{A \subset V : |A| = k\}$ that minimizes $\max_{v \in V} d(v, A)$. We can consider the semi-bandit setting where the set of base arms is $E$, $\boldsymbol{\mu}^* \triangleq (d(v_i, v_j))_{(v_i, v_j) \in E}$ and the feedback set $S$ includes the edges of the graph induced by the chosen action. We can target an approximation regret with $\alpha = 1/2$ using the following oracle (which is simply the standard greedy algorithm for this problem).

- Oracle$_1(\boldsymbol{\mu})$ : For $i \in [k]$, let $\mathcal{E}_i \triangleq \{(a_1, \ldots, a_i) : (a_1, \ldots, a_{i-1}) = E_{i-1}, a_i \in V \backslash E_{i-1}\}$ and $r_i((a_1, \ldots, a_i), \boldsymbol{\mu}) \triangleq \min_{j \in [i-1]} \mu_{a_i, a_j}$ (with $r_1 = 0$). One can notice we thus have $E_i^* \in \arg\max_{(a_1, \ldots, a_i) \in \mathcal{E}_i} d(a_i, \{a_1, \ldots, a_{i-1}\}) = \arg\max_{(a_1, \ldots, a_i) \in \mathcal{E}_i} d(a_i, E_{i-1})$.

- Oracle$_2(E_1, \ldots, E_k)$ : Let $(a_1, \ldots, a_k) \triangleq E_k$. Oracle$_2$ returns $A = \{a_1, \ldots, a_k\}$. Let $w \in \arg\max_{v \in V} d(v, A)$.

We thus have:

$$\frac{1}{2} \max_{v \in V} d(v, A) - \max_{v \in V} d(v, A^*) \leq \frac{1}{2} \left( \max_{v \in V} d(v, A) - \min_{a \in A \cup \{w\}} \min_{a' \in A \backslash \{a\}} d(a, a') \right)$$

$$= \frac{1}{2} \max_{j \in [k-1]} \left( \max_{v \in V} d(v, A) - d(a_{j+1}, E_j) \right)$$

$$\leq \frac{1}{2} \max_{j \in [k-1]} \left( \max_{v \in V} d(v, E_j) - d(a_{j+1}, E_j) \right)$$

$$= \frac{1}{2} \max_{j \in [k-1]} \left( r_{j+1}(E_{j+1}^*, \boldsymbol{\mu}^*) - r_{j+1}(E_{j+1}, \boldsymbol{\mu}^*) \right).$$

Where the first inequality is deduced as follows: the map $f : v \mapsto \arg\min_{a^* \in A^*} d(a^*, v)$ defines a partition of $V$ into $k = |A^*|$ clusters. By the the pigeonhole principle, one cluster contains 2 different points $a, a' \in A \cup \{w\}$ (simply because its size is $k + 1$). We can assume $a' \in A$ without loss of generality. Thus, since $f(a) = f(a')$, we get $d(a, a') \leq d(a, f(a)) + d(a', f(a')) = d(a, A^*) + d(a', A^*) \leq 2 \max_{v \in V} d(v, A^*)$.

**Vertex cover** The problem consists, given an undirected graph $G = (V, E)$, in finding a set of vertices with minimal cost to cover all the edges of $E$. Formally, with $\boldsymbol{\mu}^* \in \mathbb{R}_+^V$, the goal is to find an action $A \in \mathcal{A} \triangleq \{A \subset V : \forall (u, v) \in E, u \in A \text{ or } v \in A\}$ that minimizes $\mathbf{e}_A^\top \boldsymbol{\mu}^*$. The semi-bandit feedback is defined directly as $S = A$. We can target an approximation regret with $\alpha = 1/2$ using the following linear programming (LP) relaxation oracle.

- Oracle$_1(\boldsymbol{\mu})$ : Let[6] $\mathcal{E}_1 \triangleq \left\{ \mathbf{x} \in \{0, 1/2, 1\}^V : \forall (u, v) \in E, \ x_u + x_v \geq 1 \right\}$ and $r_1(\mathbf{x}, \boldsymbol{\mu}) \triangleq -\mathbf{x}^\top \boldsymbol{\mu}$.

- Oracle$_2(E_1)$ : Let $\mathbf{x} \triangleq E_1$. Oracle$_2$ returns $A = \{v \in V : x_v \geq 1/2\} \in \mathcal{A}$ (since $\forall (u, v) \in E, x_u + x_v \geq 1$, so $x_u \geq 1/2$ or $x_v \geq 1/2$, so $u \in A$ or $v \in A$). Let $\mathbf{x}^* \triangleq E_1^*$.

We have:

$$\frac{1}{2} \mathbf{e}_A^\top \boldsymbol{\mu}^* - \mathbf{e}_{A^*}^\top \boldsymbol{\mu}^* \leq \mathbf{x}^\top \boldsymbol{\mu}^* - \mathbf{e}_{A^*}^\top \boldsymbol{\mu}^* \leq \mathbf{x}^\top \boldsymbol{\mu}^* - \mathbf{x}^{*\top} \boldsymbol{\mu}^* = r_1(E_1^*, \boldsymbol{\mu}^*) - r_1(E_1, \boldsymbol{\mu}^*).$$

---

[6]The LP relaxation of vertex cover is half-integral, so that we can allow each variable to be in $\{0, 1/2, 1\}$ rather than the interval from 0 to 1.

Notice, $r_1$ satisfies Assumption 2 with the constants being 1, using $\mathbf{x} \in \{0, 1/2, 1\}^V$, so that $\mathbf{x} \leq \mathbf{e}_A$.

**Max-Cut** Given an undirected weighted graph $G = (V, E)$, with weights $\boldsymbol{\mu}^* \triangleq (\mu^*_{(u,v)})_{(u,v) \in E}$, the goal is to find an action $A \in \mathcal{A} \triangleq \{A \subset V\}$ maximizing the total weight of the edges between $A$ and its complement, i.e., $\frac{1}{2} \sum_{(u,v) \in E} \mu^*_{(u,v)}(1 - y_u y_v)$, where $\mathbf{y} \triangleq \mathbf{e}_A - \mathbf{e}_{V \setminus A}$. We consider the semi-bandit context where $E$ is the set of arms, and the feedback set includes the edges between $A$ and its complement, i.e., $S = \{(u,v) \in E, \ y_u y_v = -1\}$. To include randomization within the oracle, we can extend the action space $\mathcal{A}$ to the set of probability measures on $\{A \subset V\}$, replacing $\Delta(A)$ by its expectation on $A$, as a function of the distribution of $A$. The polynomial-time approximation algorithm for Max-Cut with the best known approximation ratio [Goemans and Williamson, 1995] uses semidefinite programming and randomized rounding, and achieves an approximation ratio of $\alpha = \frac{2}{\pi} \min_{0 \leq \theta \leq \pi} \frac{\theta}{1 - \cos \theta} \approx 0.878$. In our context, it can be defined as follows.

- $\text{Oracle}_1(\boldsymbol{\mu})$ : We define $\mathcal{E}_1 \triangleq \left\{ (\mathbf{v}_u)_{u \in V} \in \left\{ \mathbf{v} \in \mathbb{R}^V : \|\mathbf{v}\|_2 = 1 \right\}^V \right\}$ and $r_1((\mathbf{v}_u)_{u \in V}, \boldsymbol{\mu}) \triangleq \frac{1}{2} \sum_{(u,v) \in E} \mu_{(u,v)}(1 - \mathbf{v}_u^\intercal \mathbf{v}_v)$.

- $\text{Oracle}_2(E_1)$ : Let $(\mathbf{v}_u)_{u \in V} \triangleq E_1$. $\text{Oracle}_2$ returns the distribution of $A = \{u \in V, \ \mathbf{v}_u^\intercal \mathbf{Z} \geq 0\}$, where $\mathbf{Z} \sim \mathcal{U}(\{\mathbf{v} \in \mathbb{R}^V : \|\mathbf{v}\|_2 = 1\})$.

The following is proved by Goemans and Williamson [1995]:

$$
\sum_{(u,v) \in E} \mu_{(u,v)} \mathbb{P}[(u,v) \in S] = \mathbb{E}\left[ \sum_{(u,v) \in E} \mu_{(u,v)} \frac{1 - \text{sign}(\mathbf{v}_u^\intercal \mathbf{Z})\text{sign}(\mathbf{v}_v^\intercal \mathbf{Z})}{2} \right]
$$

$$
= \sum_{(u,v) \in E} \mu_{(u,v)} \frac{\arccos(\mathbf{v}_u^\intercal \mathbf{v}_v)}{\pi}
$$

$$
\geq \alpha \sum_{(u,v) \in E} \mu_{(u,v)} \frac{1 - \mathbf{v}_u^\intercal \mathbf{v}_v}{2} = \alpha r_1((\mathbf{v}_u)_{u \in V}, \boldsymbol{\mu}).
$$

Thus, $r_1$ satisfies Assumption 2 with the constants being $1/\alpha$. We also get, with $\mathbf{y}^* \triangleq \mathbf{e}_{A^*} - \mathbf{e}_{V \setminus A^*}$:

$$
\Delta(\text{Oracle}_2(E_1)) = \alpha \sum_{(u,v) \in E} \mu^*_{(u,v)} \frac{1 - y_u^* y_v^*}{2} - \mathbb{E}\left[ \sum_{(u,v) \in E} \mu^*_{(u,v)} \frac{1 - \text{sign}(\mathbf{v}_u^\intercal \mathbf{Z})\text{sign}(\mathbf{v}_v^\intercal \mathbf{Z})}{2} \right]
$$

$$
\leq \alpha(r_1(E_1^*, \boldsymbol{\mu}^*) - r_1(E_1, \boldsymbol{\mu}^*)).
$$

## 5 Conclusion

In this article, our main objective is to further expand the "approximation regret scope" of the CTS policy. We not only expand it to probabilistically triggered arms (which was one of the open questions by Kong et al. [2021]), but we also consider a broader class of oracles compatible with CTS. More precisely, we propose a condition, REDUCE2EXACT, which may seem unnatural at first, but which in fact simply expresses that sub-problems that can be solved exactly must be hidden in the original approximation problem, and that the approximation oracle exploit them to output the final solution. Knowing that the majority of approximation algorithms use one or more relaxations to an exact problem (e.g., solving a convex programming relaxation to obtain a fractional solution and then rounding this fractional solution to get a feasible solution), our assumption falls within the range of many CMAB-T settings. From this reduction, we naturally obtain the standard tight regret bound $\mathcal{O}(\log(T)/\Delta_{\min})$. This is the first tight bound for the approximation regret on non-exact oracles.

As future work, it may be interesting to explore other CMAB-T problems where the REDUCE2EXACT condition does (or doesn't) hold. We also think our setting should be generalizable to the *budgeted regret* setting without much difficulty (see Perrault et al. [2019b, 2020c,b] for examples with an approximation oracle). Finally, we have that the approximation regret is in some way conservative compared to the greedy regret. For a given oracle, we can easily consider the equivalent of the greedy regret for that oracle. An interesting investigation would then be to extend the work of Kong et al. [2021] in this direction, considering other types of oracle.

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
