# A  Proof of Theorem 1

In the following proof, we will treat both algorithms at the same time, detailing the steps where there is a difference. Thus, for Algorithm 1, we use the convention $\beta = 1$ in the following. We let $B \triangleq \|\mathbf{B}\|_\infty$ and $0 < \varepsilon < \Delta_{\min}/(2B(m^{*2}+1))$. For the two algorithms, we consider the following events for any time step $t \in \mathbb{N}^*$:

- $\mathfrak{Z}_t \triangleq \{\Delta_t > 0\}$,
- $\mathfrak{B}_t \triangleq \left\{ \left\| \boldsymbol{p}(A_t) \odot \mathbf{B} \odot (\overline{\boldsymbol{\mu}}_{t-1} - \boldsymbol{\mu}^*) \right\|_1 > \Delta_{\min}/2 - B(m^{*2}+1)\varepsilon \right\}$,
- $\mathfrak{C}_t \triangleq \left\{ \left\| \boldsymbol{p}(A_t) \odot \mathbf{B} \odot (\boldsymbol{\theta}_t - \boldsymbol{\mu}^*) \right\|_1 > \Delta_t - B(m^{*2}+1)\varepsilon \right\}$,
- $\mathfrak{D}_t \triangleq \left\{ \left\| \boldsymbol{p}(A_t) \odot \mathbf{B} \odot (\boldsymbol{\theta}_t - \overline{\boldsymbol{\mu}}_{t-1}) \right\|_1 \geq \sqrt{\sum_{i \in [n]} \frac{2\log(2^n(1+\lceil\log_2(1/p^*)\rceil)^n T)p_i(A_t)^2 B_i^2 \beta}{N_{i,t-1}}} \right\}$.

We decompose the regret analysis into several steps, each step corresponding to a filtration of the regret against a combination of these events.

**Step 1: bound under $\mathfrak{Z}_t \wedge \neg\mathfrak{C}_t$**  The filtered regret bound

$$\mathbb{E}\left[ \sum_{t=1}^T \Delta_t \mathbb{I}\{\mathfrak{Z}_t \wedge \neg\mathfrak{C}_t\} \right] \leq \Delta_{\max} \frac{cm^*}{p^*\varepsilon^2} \left( \frac{c'}{\varepsilon^4} \right)^{m^*}$$

is deduced from the following two lemmas, considering the following events for a subset $Z \subset [n]$:

$$\mathfrak{R}(\boldsymbol{\theta}', Z) \triangleq$$
$$\left\{ Z \subset \mathrm{T}(A) \text{ s.t. } A = \mathrm{Oracle}(\boldsymbol{\theta}'), \left\| \boldsymbol{p}(A) \odot \mathbf{B} \odot (\boldsymbol{\theta}' - \boldsymbol{\mu}^*) \right\|_1 > \Delta(A) - B(m^{*2}+1)\varepsilon \right\},$$

$$\mathfrak{S}_t(Z) \triangleq \left\{ \forall \boldsymbol{\theta}' \text{ s.t. } \left\| (\boldsymbol{\mu}^* - \boldsymbol{\theta}') \odot \mathbf{e}_Z \right\|_\infty \leq \varepsilon, \mathfrak{R}(\boldsymbol{\theta}' \odot \mathbf{e}_Z + \boldsymbol{\theta}_t \odot \mathbf{e}_{Z^c}, Z) \text{ holds} \right\},$$
$$\mathfrak{T}_t(Z) \triangleq \left\{ \left\| (\boldsymbol{\mu}^* - \boldsymbol{\theta}_t) \odot \mathbf{e}_Z \right\|_\infty > \varepsilon \right\}.$$

**Lemma 1.**

$$\mathfrak{Z}_t, \neg\mathfrak{C}_t \Rightarrow \exists Z \subset \mathrm{T}(A^*), Z \neq \emptyset \text{ s.t. the event } \mathfrak{S}_t(Z) \wedge \mathfrak{T}_t(Z) \text{ holds.}$$

**Lemma 2.** *There are two constants $c, c'$ such that*

$$\sum_{Z \subset \mathrm{T}(A^*), Z \neq \emptyset} \mathbb{E}\left[ \sum_{t=1}^T \mathbb{I}\{\mathfrak{S}_t(Z), \mathfrak{T}_t(Z)\} \right] \leq \frac{cm^*}{p^*\varepsilon^2} \left( \frac{c'}{\varepsilon^4} \right)^{m^*}.$$

*Proof of Lemma 1.*  It is sufficient to prove that

$$\mathfrak{Z}_t, \neg\mathfrak{C}_t \Rightarrow \exists Z \subset \mathrm{T}(A^*), Z \neq \emptyset \text{ s.t. } \mathfrak{S}_t(Z) \text{ holds,} \tag{4}$$

because $\neg\mathfrak{C}_t$ and $\mathfrak{S}_t(Z)$ together imply $\mathfrak{T}_t(Z)$. Then, we get (4) in a similar way as Lemma 2 from Huyuk and Tekin [2019], which is possible as their Assumption 3 is implied by our Assumption 2. More precisely, the only places where we use our Assumption 2 instead of their Assumption 3 are in cases $1a, 2a, \dots$, when we show that $\mathfrak{R}(\boldsymbol{\theta}' \odot \mathbf{e}_Z + \boldsymbol{\theta}_t \odot \mathbf{e}_{Z^c}, Z)$ holds. The detailed proof is given in the following.

We first consider the choice $Z = Z_1 = \mathrm{T}(A^*)$. Two cases can be distinguished:

1a) $\forall \boldsymbol{\theta}'$ s.t. $\left\| (\boldsymbol{\mu}^* - \boldsymbol{\theta}') \odot \mathbf{e}_{Z_1} \right\|_\infty \leq \varepsilon$, we have $Z_1 \subset \mathrm{T}\big(\mathrm{Oracle}(\boldsymbol{\theta}' \odot \mathbf{e}_{Z_1} + \boldsymbol{\theta}_t \odot \mathbf{e}_{Z_1^c})\big)$.

1b) $\exists \boldsymbol{\theta}'$ s.t. $\left\| (\boldsymbol{\mu}^* - \boldsymbol{\theta}') \odot \mathbf{e}_{Z_1} \right\|_\infty \leq \varepsilon$ such that $Z_1 \not\subset \mathrm{T}\big(\mathrm{Oracle}(\boldsymbol{\theta}' \odot \mathbf{e}_{Z_1} + \boldsymbol{\theta}_t \odot \mathbf{e}_{Z_1^c})\big)$.

**1a)** For the first case, consider any vector $\boldsymbol{\theta}'$ such that $\left\| (\boldsymbol{\mu}^* - \boldsymbol{\theta}') \odot \mathbf{e}_{Z_1} \right\|_\infty \overset{(5)}{\leq} \varepsilon$ and let $A \overset{(6)}{=} \mathrm{Oracle}(\boldsymbol{\theta}' \odot \mathbf{e}_{Z_1} + \boldsymbol{\theta}_t \odot \mathbf{e}_{Z_1^c})$. We can write

$$r\big(A, \boldsymbol{\theta}' \odot \mathbf{e}_{Z_1} + \boldsymbol{\theta}_t \odot \mathbf{e}_{Z_1^c}\big) \overset{(7)}{\geq} r\big(A^*, \boldsymbol{\theta}' \odot \mathbf{e}_{Z_1} + \boldsymbol{\theta}_t \odot \mathbf{e}_{Z_1^c}\big) \overset{(8)}{\geq} r(A^*, \boldsymbol{\mu}^*) - Bm^*\varepsilon,$$

where (7) is from (6), and (8) is from (5). This rewrites as

$$r\big(A, \boldsymbol{\theta}' \odot \mathbf{e}_{Z_1} + \boldsymbol{\theta}_t \odot \mathbf{e}_{Z_1^c}\big) \geq r(A^*, \boldsymbol{\mu}^*) - Bm^*\varepsilon > r(A^*, \boldsymbol{\mu}^*) - B\Big(m^{*2} + 1\Big)\varepsilon,$$

so $\mathfrak{R}_t(\boldsymbol{\theta}' \odot \mathbf{e}_{Z_1} + \boldsymbol{\theta}_t \odot \mathbf{e}_{Z_1^c}, Z_1)$ holds. Therefore, we have proved that $\mathfrak{S}_t(Z_1)$ holds.

**1b)** For the second case, we have some vector $\boldsymbol{\theta}'$ such that $\big\|(\boldsymbol{\mu}^* - \boldsymbol{\theta}') \odot \mathbf{e}_{Z_1}\big\|_\infty \overset{(9)}{\leq} \varepsilon$, and some action $A = \mathrm{Oracle}\big(\boldsymbol{\theta}' \odot \mathbf{e}_{Z_1} + \boldsymbol{\theta}_t \odot \mathbf{e}_{Z_1^c}\big)$ such that $Z_1 \not\subset \mathrm{T}(A)$. We consider $Z_2 = Z_1 \cap \mathrm{T}(A) \neq Z_1$. We first prove that $Z_2 \neq \emptyset$ by showing that if an action $A'$ is such that $Z_1 \cap \mathrm{T}(A') \overset{(10)}{=} \emptyset$, then $A' \neq A$ as it has a lower reward value than that of $A^*$:

$$\begin{aligned}
r\big(A', \boldsymbol{\theta}' \odot \mathbf{e}_{Z_1} + \boldsymbol{\theta}_t \odot \mathbf{e}_{Z_1^c}\big) &\overset{(11)}{=} r(A', \boldsymbol{\theta}_t) \overset{(12)}{\leq} r(A_t, \boldsymbol{\theta}_t) \\
&\overset{(13)}{\leq} r(A^*, \boldsymbol{\mu}^*) - B\Big(m^{*2} + 1\Big)\varepsilon \\
&< r(A^*, \boldsymbol{\mu}^*) - Bm^*\varepsilon \\
&\overset{(14)}{\leq} r\big(A^*, \boldsymbol{\theta}' \odot \mathbf{e}_{Z_1} + \boldsymbol{\theta}_t \odot \mathbf{e}_{Z_1^c}\big),
\end{aligned}$$

where (11) is from (10), (12) is from the definition of $A_t$, (13) is from $\neg\mathfrak{C}_t$ and (14) is from (9). Now, we again distinguish two cases:

    2a) $\forall \boldsymbol{\theta}''$ s.t. $\big\|(\boldsymbol{\mu}^* - \boldsymbol{\theta}'') \odot \mathbf{e}_{Z_2}\big\|_\infty \leq \varepsilon$, we have $Z_2 \subset \mathrm{T}\big(\mathrm{Oracle}(\boldsymbol{\theta}'' \odot \mathbf{e}_{Z_2} + \boldsymbol{\theta}_t \odot \mathbf{e}_{Z_2^c})\big)$.

    2b) $\exists \boldsymbol{\theta}''$ s.t. $\big\|(\boldsymbol{\mu}^* - \boldsymbol{\theta}'') \odot \mathbf{e}_{Z_2}\big\|_\infty \leq \varepsilon$ such that $Z_2 \not\subset \mathrm{T}\big(\mathrm{Oracle}(\boldsymbol{\theta}' \odot \mathbf{e}_{Z_2} + \boldsymbol{\theta}_t \odot \mathbf{e}_{Z_2^c})\big)$.

Notice that when $\big\|(\boldsymbol{\mu}^* - \boldsymbol{\theta}'') \odot \mathbf{e}_{Z_2}\big\|_\infty \overset{(15)}{\leq} \varepsilon$, then

$$r\big(A, \boldsymbol{\theta}'' \odot \mathbf{e}_{Z_2} + \boldsymbol{\theta}_t \odot \mathbf{e}_{Z_2^c}\big) \geq r\big(A, \boldsymbol{\theta}' \odot \mathbf{e}_{Z_1} + \boldsymbol{\theta}_t \odot \mathbf{e}_{Z_1^c}\big) - 2B(m^* - 1)\varepsilon. \tag{16}$$

Indeed, (16) is a consequence of

$$\begin{aligned}
&\big\|\big(\boldsymbol{\theta}' \odot \mathbf{e}_{Z_1} + \boldsymbol{\theta}_t \odot \mathbf{e}_{Z_1^c} - \boldsymbol{\theta}'' \odot \mathbf{e}_{Z_2} - \boldsymbol{\theta}_t \odot \mathbf{e}_{Z_2^c}\big) \odot \mathbf{e}_{\mathrm{T}(A)}\big\|_1 \\
&= \big\|(\boldsymbol{\theta}' - \boldsymbol{\theta}'') \odot \mathbf{e}_{Z_2}\big\|_1 \\
&\leq \big\|(\boldsymbol{\mu}^* - \boldsymbol{\theta}') \odot \mathbf{e}_{Z_2}\big\|_1 + \big\|(\boldsymbol{\mu}^* - \boldsymbol{\theta}'') \odot \mathbf{e}_{Z_2}\big\|_1 \\
&\leq 2(m^* - 1)\varepsilon,
\end{aligned}$$

where we used (15), (9) and that $Z_2$ is strictly included in $Z_1$.

**2a)** For the first case, considering any vector $\boldsymbol{\theta}''$ such that $\big\|(\boldsymbol{\mu}^* - \boldsymbol{\theta}'') \odot \mathbf{e}_{Z_2}\big\|_\infty \leq \varepsilon$, we have with $\widetilde{A} = \mathrm{Oracle}\big(\boldsymbol{\theta}'' \odot \mathbf{e}_{Z_2} + \boldsymbol{\theta}_t \odot \mathbf{e}_{Z_2^c}\big)$ that

$$\begin{aligned}
r\Big(\widetilde{A}, \boldsymbol{\theta}'' \odot \mathbf{e}_{Z_2} + \boldsymbol{\theta}_t \odot \mathbf{e}_{Z_2^c}\Big) &\geq r\big(A, \boldsymbol{\theta}'' \odot \mathbf{e}_{Z_2} + \boldsymbol{\theta}_t \odot \mathbf{e}_{Z_2^c}\big) \\
&\overset{(17)}{\geq} r\big(A, \boldsymbol{\theta}' \odot \mathbf{e}_{Z_1} + \boldsymbol{\theta}_t \odot \mathbf{e}_{Z_1^c}\big) - 2B(m^* - 1)\varepsilon \\
&\geq r\big(A^*, \boldsymbol{\theta}' \odot \mathbf{e}_{Z_1} + \boldsymbol{\theta}_t \odot \mathbf{e}_{Z_1^c}\big) - 2B(m^* - 1)\varepsilon \\
&\overset{(18)}{\geq} r(A^*, \boldsymbol{\mu}^*) - Bm^*\varepsilon - 2B(m^* - 1)\varepsilon \\
&\geq r(A^*, \boldsymbol{\mu}^*) - B(m^{*2} + 1)\varepsilon,
\end{aligned}$$

where (17) uses (16) and (18) uses (9). Therefore, $\mathfrak{R}_t(\boldsymbol{\theta}' \odot \mathbf{e}_{Z_2} + \boldsymbol{\theta}_t \odot \mathbf{e}_{Z_2^c}, Z_2)$ holds, and we proved that $\mathfrak{S}_t(Z_2)$ holds.

**2b)** For the second case, we have a vector $\boldsymbol{\theta}''$ such that $\big\|(\boldsymbol{\mu}^* - \boldsymbol{\theta}'') \odot \mathbf{e}_{Z_2}\big\|_\infty \leq \varepsilon$ and an action $\widetilde{A} = \mathrm{Oracle}\big(\boldsymbol{\theta}'' \odot \mathbf{e}_{Z_2} + \boldsymbol{\theta}_t \odot \mathbf{e}_{Z_2^c}\big)$ such that $Z_2 \not\subset \mathrm{T}\big(\widetilde{A}\big)$. We consider $Z_3 =$

$Z_2 \cap \mathrm{T}\!\left(\widetilde{A}\right)$. Again, $Z_3 \neq \emptyset$ because for any $A''$ such that $\mathrm{T}(A'') \cap Z_2 = \emptyset$, we have $A'' \neq \mathrm{Oracle}\!\left(\boldsymbol{\theta}'' \odot \mathbf{e}_{Z_2} + \boldsymbol{\theta}_t \odot \mathbf{e}_{Z_2{}^c}\right)$:

$$
\begin{aligned}
r\!\left(A'', \boldsymbol{\theta}'' \odot \mathbf{e}_{Z_2} + \boldsymbol{\theta}_t \odot \mathbf{e}_{Z_2{}^c}\right) = r(A'', \boldsymbol{\theta}_t) &\leq r(A_t, \boldsymbol{\theta}_t) \\
&\leq r(A^*, \boldsymbol{\mu}^*) - B\!\left(m^{*2} + 1\right)\varepsilon \\
&< r(A^*, \boldsymbol{\mu}^*) - B m^* \varepsilon - 2B(m^* - 1)\varepsilon \\
&\leq r\!\left(A, \boldsymbol{\theta}'' \odot \mathbf{e}_{Z_2} + \boldsymbol{\theta}_t \odot \mathbf{e}_{Z_2{}^c}\right),
\end{aligned}
$$

where the last inequality is obtained in the same way as in inequalities from (17) to (18).

We could repeat the above argument and each time the size $Z_i$ is decreased by at least 1. Thus, after at most $m^* - 1$ steps, since $m^* + 2(m^* - 1) + 2(m^* - 2) + \cdots + 2 = m^{*2} < m^{*2} + 1$, we could reach the end and find a $Z_i \neq \emptyset$ such that $\mathfrak{S}_t(Z_i)$ holds. $\qquad\square$

*Proof of Lemma 2.* Let $Z = \left\{z_0, \ldots, z_{|Z|-1}\right\} \subset \mathrm{T}(A^*)$, $Z \neq \emptyset$. If $|Z| = 1$, we let

$$
\eta_{q,0} \triangleq \{t \geq 1,\ |\{t' \in [t-1],\ \mathfrak{S}_{t'}(Z) \wedge \neg \mathfrak{T}_{t'}(Z) \wedge \{z_0 \in S_{t'}\}\}| = q\}.
$$

Else, for $t > 1$, we recursively define

$$
c_{t+1} \triangleq c_t + \mathbb{I}\{\mathfrak{S}_t(Z) \wedge \neg \mathfrak{T}_t(Z) \wedge \{z_{c_t} \in S_t\}\} \mod |Z|,
$$

with $c_1 \triangleq 0$ and let

$$
\eta_{q,k} \triangleq \{t \geq 1,\ c_t = k,\ |t' \in [t-1],\ c_{t'} = k \neq c_{t'+1}| = q\}.
$$

Notice that for $\tau \geq \inf \eta_{q,0}$, we have $N_{i,\tau-1} \geq q$ for all $i \in Z$. We have

$$
\begin{aligned}
\mathbb{E}\!\left[\sum_{t=1}^{T} \mathbb{I}\{\mathfrak{S}_t(Z), \mathfrak{T}_t(Z)\}\right] &= \sum_{q \geq 0} \sum_{k=0}^{|Z|-1} \mathbb{E}\!\left[\sum_{t \in \eta_{q,k}} \mathbb{I}\{\mathfrak{S}_t(Z), \mathfrak{T}_t(Z)\}\right] \\
&\leq \frac{1}{p^*} \sum_{q \geq 0} \sum_{k=0}^{|Z|-1} \mathbb{E}\!\left[\sum_{t \in \eta_{q,k}} \mathbb{I}\{\mathfrak{S}_t(Z), \mathfrak{T}_t(Z), z_{c_t} \in S_t\}\right] \\
&\leq \frac{1}{p^*} \sum_{q \geq 0} \sum_{k=0}^{|Z|-1} \left(\mathbb{E}\!\left[\sup_{\tau \geq \inf \eta_{q,k}} \frac{1}{\mathbb{P}[\neg \mathfrak{T}_\tau(Z)|\mathcal{H}_\tau]}\right] - 1\right) \\
&\leq \frac{|Z|}{p^*} \sum_{q \geq 0} \left(\mathbb{E}\!\left[\sup_{\tau \geq \inf \eta_{q,0}} \prod_{i \in Z} \frac{1}{\mathbb{P}[|\theta_{i,\tau} - \mu_i^*| \leq \varepsilon|\mathcal{H}_\tau]}\right] - 1\right).
\end{aligned}
$$

From the initialization phase, we can assume that the event

$$
\mathfrak{M}_t \triangleq \{\forall i \in [n],\ N_{i,t-1} \geq 1\}
$$

holds (under the complementary event, we have the upper bound $n$). If there is no initialization, we can have $q = 0$ in the following, noticing that when $\theta_{i,t}$ is uniform on $[a, b]$, then the probability $\mathbb{P}[|\theta_{i,t} - \mu_i^*| \leq \varepsilon|\mathcal{H}_t]$ is equal to $2\varepsilon/(b - a)$. We are thus interested in bounding

$$
\frac{|Z|}{p^*} \underbrace{\sum_{q \geq 1} \left(\mathbb{E}\!\left[\sup_{\tau \geq \tau_q} \prod_{i \in Z} \frac{1}{\mathbb{P}[|\theta_{i,\tau} - \mu_i^*| \leq \varepsilon|\mathcal{H}_\tau]}\right] - 1\right)}_{(19)},
$$

with $\tau_q \triangleq \inf \eta_{q,0}$. We have

$$
(19) \leq \sum_{q \geq 1} \mathbb{E}\!\left[\sup_{\tau \geq \tau_q} \sum_{Z' \subset Z,\ Z' \neq \emptyset} \prod_{i \in Z'} \left(\frac{1}{\mathbb{P}[|\theta_{i,\tau} - \mu_i^*| \leq \varepsilon|\mathcal{H}_\tau]} - 1\right)\right]
$$

$$\leq \sum_{q \geq 1} \sum_{Z' \subset Z, \ Z' \neq \emptyset} \underbrace{\mathbb{E}\left[\sup_{\tau \geq \tau_q} \prod_{i \in Z'} \left(\frac{1}{\mathbb{P}[\,|\theta_{i,\tau} - \mu_i^*| \leq \varepsilon | \mathcal{H}_\tau]} - 1\right)\right]}_{(20)}$$

Then, we can take a union bound on the counters:

$$(20) \leq \sum_{\mathbf{k} \in [q..\infty)^{Z'}} \mathbb{E}\left[\sup_{\tau \geq \tau_q} \mathbb{I}\{\forall i \in Z', \ N_{i,\tau-1} = k_i\} \prod_{i \in Z'} \left(\frac{1}{\mathbb{P}[\,|\theta_{i,\tau} - \mu_i^*| \leq \varepsilon | \mathcal{H}_\tau]} - 1\right)\right].$$

From this point, there are two distinct analysis depending on whether we consider Algorithm 1 or Algorithm 2.

**For Algorithm 1:**

For any arm $i \in [n]$, $k_i \in \mathbb{N}$, we define $p_{i,k_i}$ as the probability of $\left|\widetilde{\theta}_{i,k_i} - \mu_i^*\right| \leq \varepsilon$, where $\widetilde{\theta}_{i,k_i}$ is a sample from the posterior of arm $i$ when there are $k_i$ observations of arm $i$ (i.e., $p_{i,k_i}$ is a random variable measurable with respect to those $k_i$ independent draws of arm $i$). We have

$$\mathbb{E}\left[\sup_{\tau \geq \tau_q} \mathbb{I}\{\forall i \in Z', \ N_{i,\tau-1} = k_i\} \prod_{i \in Z'} \left(\frac{1}{\mathbb{P}[\,|\theta_{i,\tau} - \mu_i^*| \leq \varepsilon | \mathcal{H}_\tau]} - 1\right)\right] = \mathbb{E}\left[\prod_{i \in Z'} \left(\frac{1}{p_{i,k_i}} - 1\right)\right],$$

$$= \prod_{i \in Z'} \mathbb{E}\left[\left(\frac{1}{p_{i,k_i}} - 1\right)\right].$$

From Lemma 5,6 in Wang and Chen [2018], we know that

$$\mathbb{E}\left[\frac{1}{p_{i,k_i}}\right] \leq \begin{cases} 4/\varepsilon^2 & \text{for every } k_i \geq 0 \\ 1 + 6c'' \cdot e^{-\varepsilon^2 k_i/2}\varepsilon^{-2} + \frac{2}{e^{\varepsilon^2 k_i/8} - 2} & \text{if } k_i > 8/\varepsilon^2, \end{cases}$$

for some universal constant $c''$. There are thus two cases: If $q > 8/\varepsilon^2$, then some simple calculations show that $\sum_{Z' \subset Z, \ Z' \neq \emptyset} (20)$ is bounded by a term of the form $e^{-\varepsilon^2 q/8}\left(c'\varepsilon^{-4}\right)^{|Z|}$, where $c'$ is a universal constant, and if $q \leq 8/\varepsilon^2$, then $\sum_{Z' \subset Z, \ Z' \neq \emptyset} (20)$ is bounded by $\left(c\varepsilon^{-4}\right)^{|Z|}$, where $c$ is a universal constant. Summing over $q \geq 1$, we thus get the desired result.

**For Algorithm 2:**

One can notice that for all $i \in Z'$, all $k_i \geq q$, $\mathbb{I}\{N_{i,\tau-1} = k_i\}\left(\frac{1}{\mathbb{P}[\,|\theta_{i,\tau} - \mu_i^*| \leq \varepsilon | \mathcal{H}_\tau]} - 1\right)$ is of the form $\mathbb{I}\{N_{i,\tau-1} = k_i\}g_i\left(\left|\overline{\mu}_{i,\tau-1} - \mu_i^*\right|\right)$, with $g_i$ being an increasing function on $\mathbb{R}_+$. Indeed, we see that the conditional distribution of $\theta_{i,\tau} - \overline{\mu}_{i,\tau-1}$ is $\mathcal{N}\left(0, \beta N_{i,\tau-1}^{-1}/4\right)$, which is symmetric, so we have

$$\mathbb{P}[\,|\theta_{i,\tau} - \mu_i^*| \leq \varepsilon | \mathcal{H}_\tau] = \mathbb{P}\big[\,\big|\theta_{i,\tau} - \overline{\mu}_{i,\tau-1} + \big|\overline{\mu}_{i,\tau-1} - \mu_i^*\big|\big| \leq \varepsilon \big| \mathcal{H}_\tau\big].$$

In addition, under $\mathbb{I}\{N_{i,\tau-1} = k_i\}$, the conditional distribution of $\theta_{i,\tau} - \overline{\mu}_{i,\tau-1}$ does not depend on the history, but only on $k_i$. Therefore, the above probability is a function of $\left|\overline{\mu}_{i,\tau-1} - \mu_i^*\right|$ and so the function $g_i$ exists. It is increasing on $\mathbb{R}_+$ because for any fixed $\sigma > 0$,

$$\frac{\partial}{\partial x}\int_{x-\varepsilon}^{x+\varepsilon} \frac{1}{\sqrt{2\pi\sigma^2}} e^{-\frac{u^2}{2\sigma^2}}\mathrm{d}u = \frac{1}{\sqrt{2\pi\sigma^2}}\left(e^{-\frac{(x+\varepsilon)^2}{2\sigma^2}} - e^{-\frac{(x-\varepsilon)^2}{2\sigma^2}}\right) < 0 \text{ for } x > 0.$$

In particular, we can consider the inverse function $g_i^{-1}$. We now want to use a stochastic dominance argument in order to treat the outcomes as if they were Gaussian: we have for any $\mathbf{k} \in [q..\infty)^{Z'}$,

$$\mathbb{E}\left[\sup_{\tau \geq \tau_q} \prod_{i \in Z'} \left(\mathbb{I}\{N_{i,\tau-1} = k_i\}g_i\left(\left|\overline{\mu}_{i,\tau-1} - \mu_i^*\right|\right)\right)\right]$$

$$= \mathbb{E}\left[\sup_{\tau \geq \tau_q} \prod_{i \in Z'} \left(\mathbb{I}\{N_{i,\tau-1} = k_i\}\int_0^\infty \mathbb{I}\{g_i\left(\left|\overline{\mu}_{i,\tau-1} - \mu_i^*\right|\right) \geq u_i\}\mathrm{d}u_i\right)\right]$$

$$\leq \int_{\mathbf{u}\in\mathbb{R}_+^{Z'}} \mathbb{E}\left[\sup_{\tau\geq\tau_q}\prod_{i\in Z'}\mathbb{I}\{N_{i,\tau-1}=k_i\}\mathbb{I}\{g_i(|\overline{\mu}_{i,\tau-1}-\mu_i^*|)\geq u_i\}\right]d\mathbf{u}$$

$$= \int_{\mathbf{u}\in\mathbb{R}_+^{Z'}} \mathbb{E}\left[\prod_{i\in Z'}\mathbb{I}\{N_{i,\tau^*-1}=k_i\}\mathbb{I}\{g_i(|\overline{\mu}_{i,\tau^*-1}-\mu_i^*|)\geq u_i\}\right]d\mathbf{u}, \tag{21}$$

where $\tau^*$ is the first $\tau\geq\tau_q$ such that $\mathbb{I}\{\forall i\in Z',\ N_{i,\tau-1}=k_i$ and $g_i(|\overline{\mu}_{i,\tau-1}-\mu_i^*|)\geq u_i\}$ holds, and is $\infty$ if it never holds.

$$(21) = \int_{\mathbf{u}\in\mathbb{R}_+^{Z'}} \mathbb{E}\left[\prod_{i\in Z'}\mathbb{I}\{N_{i,\tau^*-1}=k_i\}\mathbb{I}\{g_i(|\overline{\mu}_{i,\tau^*-1}-\mu_i^*|)\geq u_i\vee g_i(0)\}\right]d\mathbf{u}$$

$$= \int_{\mathbf{u}\in\mathbb{R}_+^{Z'}} \mathbb{E}\left[\prod_{i\in Z'}\mathbb{I}\{N_{i,\tau^*-1}=k_i\}\mathbb{I}\{|\overline{\mu}_{i,\tau^*-1}-\mu_i^*|\geq g_i^{-1}(u_i\vee g_i(0))\}\right]d\mathbf{u}$$

$$= \int_{\mathbf{u}\in\mathbb{R}_+^{Z'}} \sum_{\mathbf{s}\in\{-1,1\}^{Z'}} \mathbb{E}\left[\underbrace{\prod_{i\in Z'}\mathbb{I}\{N_{i,\tau^*-1}=k_i\}\mathbb{I}\{s_i(\overline{\mu}_{i,\tau^*-1}-\mu_i^*)\geq g_i^{-1}(u_i\vee g_i(0))\}}_{(22)}\right]d\mathbf{u}$$

$$(22)\leq \mathbb{P}\left[\frac{e^{\sum_{i\in Z'}N_{i,\tau^*-1}\left(4s_ig_i^{-1}(u_i\vee g_i(0))(\overline{\mu}_{i,\tau^*-1}-\mu_i^*)-2\left(g_i^{-1}(u_i\vee g_i(0))\right)^2\right)}}{e^{\sum_{i\in Z'}2\left(g_i^{-1}(u_i\vee g_i(0))\right)^2 k_i}}\geq 1, (N_{i,\tau^*-1})_{i\in Z'}=\mathbf{k}\right]$$

$$\leq \mathbb{P}\left[\frac{e^{\sum_{i\in Z'}N_{i,\tau^*-1}\left(4s_ig_i^{-1}(u_i\vee g_i(0))(\overline{\mu}_{i,\tau^*-1}-\mu_i^*)-2\left(g_i^{-1}(u_i\vee g_i(0))\right)^2\right)}}{e^{\sum_{i\in Z'}2\left(g_i^{-1}(u_i\vee g_i(0))\right)^2 k_i}}\geq 1\right]$$

$$\leq \frac{\mathbb{E}\left[\exp\left(\sum_{i\in Z'}N_{i,\tau^*-1}\left(4s_ig_i^{-1}(u_i\vee g_i(0))(\overline{\mu}_{i,\tau^*-1}-\mu_i^*)-2\left(g_i^{-1}(u_i\vee g_i(0))\right)^2\right)\right)\right]}{\exp\left(\sum_{i\in Z'}4\left(g_i^{-1}(u_i\vee g_i(0))\right)^2 k_i\right)}$$

$$= \frac{\mathbb{E}\left[\exp\left(\sum_{t=1}^{\tau^*-1}\sum_{i\in Z'\cap S_t}\left(4s_ig_i^{-1}(u_i\vee g_i(0))(X_{i,t}-\mu_i^*)-2\left(g_i^{-1}(u_i\vee g_i(0))\right)^2\right)\right)\right]}{\exp\left(\sum_{i\in Z'}2\left(g_i^{-1}(u_i\vee g_i(0))\right)^2 k_i\right)}.$$

From Assumption 3, and from the fact that either $D_{\text{trig}}$ is independent from the outcomes, or the outcomes are mutually independent and each individual outcome is independent from the fact that it is triggered, we have that

$$M_\tau = \exp\left(\sum_{t=1}^{\tau-1}\sum_{i\in Z'\cap S_t}\left(4s_ig_i^{-1}(u_i\vee g_i(0))(X_{i,t}-\mu_i^*)-2\left(g_i^{-1}(u_i\vee g_i(0))\right)^2\right)\right)$$

is a supermartingale:

$$\mathbb{E}[M_\tau|\mathcal{F}_{\tau-1}] = M_{\tau-1}\mathbb{E}\left[e^{\sum_{i\in Z'\cap S_{\tau-1}}\left(4s_ig_i^{-1}(u_i\vee g_i(0))(X_{i,\tau-1}-\mu_i^*)-2\left(g_i^{-1}(u_i\vee g_i(0))\right)^2\right)}\Big|\mathcal{F}_{\tau-1}\right]$$

$$\leq M_{\tau-1}.$$

Since $\tau^*$ is a stopping time with respect to $\mathcal{F}_\tau$, we have from Doob's optional sampling theorem for non-negative supermartingales[7] that $\mathbb{E}[M_{\tau^*}]\leq 1$. Therefore,

$$(22)\leq \exp\left(-\sum_{i\in Z'}2\left(g_i^{-1}(u_i\vee g_i(0))\right)^2 k_i\right).$$

---

[7]We use the version that relies on Fatou's lemma (Durrett [2019], Theorem 5.7.6), so that it is not needed to have any additional condition on the stopping time $\tau^*$.

Now, we want to use the following fact (see Chang et al. [2011]): if $\eta \sim \mathcal{N}(0,1)$, then with $\beta > 1$,

$$\sqrt{\frac{2e}{\pi}} \frac{\sqrt{\beta-1}}{\beta} e^{-\beta x^2/2} \leq \mathbb{P}[|\eta| \geq x].$$

Indeed, this gives

$$\sqrt{\frac{2e}{\pi}} \frac{\sqrt{\beta-1}}{\beta} \exp\left(-2\left(g_i^{-1}(u_i \vee g_i(0))\right)^2 k_i\right) \leq \mathbb{P}\left[|\eta_i| \geq g_i^{-1}(u_i \vee g_i(0))\sqrt{\frac{4k_i}{\beta}}\right],$$

where $\boldsymbol{\eta} \sim \mathcal{N}(0,1)^{\otimes Z'}$. Thus,

$$(21) \leq \left(\sqrt{\frac{\pi}{2e}} \frac{2\beta}{\sqrt{\beta-1}}\right)^{|Z'|} \int_{\mathbf{u} \in \mathbb{R}_+^{Z'}} \prod_{i \in Z'} \mathbb{P}\left[\sqrt{\frac{\beta}{4k_i}}|\eta_i| \geq g_i^{-1}(u_i \vee g_i(0))\right] d\mathbf{u}$$

$$= \left(\sqrt{\frac{\pi}{2e}} \frac{2\beta}{\sqrt{\beta-1}}\right)^{|Z'|} \int_{\mathbf{u} \in \mathbb{R}_+^{Z'}} \prod_{i \in Z'} \mathbb{P}\left[g_i\left(\sqrt{\frac{\beta}{4k_i}}|\eta_i|\right) \geq u_i \vee g_i(0)\right] d\mathbf{u}$$

$$= \left(\sqrt{\frac{\pi}{2e}} \frac{2\beta}{\sqrt{\beta-1}}\right)^{|Z'|} \int_{\mathbf{u} \in \mathbb{R}_+^{Z'}} \prod_{i \in Z'} \mathbb{P}\left[g_i\left(\sqrt{\frac{\beta}{4k_i}}|\eta_i|\right) \geq u_i\right] d\mathbf{u}$$

$$= \left(\sqrt{\frac{\pi}{2e}} \frac{2\beta}{\sqrt{\beta-1}}\right)^{|Z'|} \prod_{i \in Z'} \int_0^\infty \mathbb{P}\left[g_i\left(\sqrt{\frac{\beta}{4k_i}}|\eta_i|\right) \geq u_i\right] du_i$$

$$= \left(\sqrt{\frac{\pi}{2e}} \frac{2\beta}{\sqrt{\beta-1}}\right)^{|Z'|} \prod_{i \in Z'} \mathbb{E}\left[g_i\left(\sqrt{\frac{\beta}{4k_i}}|\eta_i|\right)\right].$$

We now want to bound $\mathbb{E}\left[g_i\left(\sqrt{\frac{\beta}{4k_i}}|\eta_i|\right)\right]$. We define $\alpha = 2 - \sqrt{2}$, the unique solution in $(1/2, 1)$ of $\alpha - 1/2 = (\alpha-1)^2/2$. Notice that $\alpha - 1/2 \geq 1/12$. Define $\varepsilon_i \triangleq \varepsilon\sqrt{\frac{4k_i}{\beta}}$. By definition, we have

$$\mathbb{E}\left[g_i\left(\sqrt{\frac{\beta}{4k_i}}|\eta_i|\right)\right] = \int_{-\infty}^{+\infty} \frac{e^{-x^2/2}}{\int_{x-\varepsilon_i}^{x+\varepsilon_i} e^{-y^2/2}dy} dx - 1$$

$$= 2\underbrace{\int_{\alpha\varepsilon_i}^{+\infty} \frac{1}{\int_{x-\varepsilon_i}^{x+\varepsilon_i} e^{-\frac{y^2-x^2}{2}}dy} dx}_{A_1} + \underbrace{\int_{-\alpha\varepsilon_i}^{\alpha\varepsilon_i} \frac{e^{-x^2/2}}{\int_{x-\varepsilon_i}^{x+\varepsilon_i} e^{-y^2/2}dy} dx - 1}_{A_2}.$$

We first bound $A_1$. With the change of variable $u = y - x$, we get:

$$A_1 = 2\int_{\alpha\varepsilon_i}^{+\infty} \frac{1}{\int_{-\varepsilon_i}^{\varepsilon_i} e^{-u^2/2-ux}du} dx$$

$$\leq 2\int_{\alpha\varepsilon_i}^{+\infty} \frac{1}{\int_{-\varepsilon_i}^0 e^{-u^2/2-ux}du} dx$$

Note that for $x \geq \alpha\varepsilon_i$ and $u \in [-\varepsilon_i, 0]$, $-u^2/2 - ux \geq -(1 - \frac{1}{2\alpha})ux$ and thus:

$$A_1 \leq 2\int_{\alpha\varepsilon_i}^{+\infty} \frac{1}{\int_{-\varepsilon_i}^0 e^{-(1-\frac{1}{2\alpha})ux}du} dx$$

$$= 2\int_{\alpha\varepsilon_i}^{+\infty} \frac{(1-\frac{1}{2\alpha})x}{e^{(1-\frac{1}{2\alpha})\varepsilon_i x} - 1} dx. \qquad (23)$$

We distinguish two regimes. First, if $\varepsilon_i^2 \geq 12$, then

$$(23) \leq \frac{2e^{(\alpha-\frac{1}{2})\varepsilon_i^2}}{e^{(\alpha-\frac{1}{2})\varepsilon_i^2} - 1} \int_{\alpha\varepsilon_i}^{+\infty} \left(1 - \frac{1}{2\alpha}\right) x e^{-(1-\frac{1}{2\alpha})\varepsilon_i x} dx$$

$$= \frac{2e^{(\alpha-\frac{1}{2})\varepsilon_i^2}}{e^{(\alpha-\frac{1}{2})\varepsilon_i^2}-1} \frac{1}{(1-\frac{1}{2\alpha})\varepsilon_i^2} \int_{(\alpha-\frac{1}{2})\varepsilon_i^2}^{+\infty} xe^{-x}\mathrm{d}x$$

$$= \frac{2e^{(\alpha-\frac{1}{2})\varepsilon_i^2}}{e^{(\alpha-\frac{1}{2})\varepsilon_i^2}-1} \frac{1}{(1-\frac{1}{2\alpha})\varepsilon_i^2} \left[-(x+1)e^{-x}\right]_{(\alpha-\frac{1}{2})\varepsilon_i^2}^{\infty}$$

$$= \frac{2e^{(\alpha-\frac{1}{2})\varepsilon_i^2}}{e^{(\alpha-\frac{1}{2})\varepsilon_i^2}-1} \frac{1}{(1-\frac{1}{2\alpha})\varepsilon_i^2} \left(\left(\alpha-\frac{1}{2}\right)\varepsilon_i^2+1\right)e^{-(\alpha-\frac{1}{2})\varepsilon_i^2}$$

$$= \frac{2}{e^{(\alpha-\frac{1}{2})\varepsilon_i^2}-1} \left(\alpha + \frac{\alpha}{(\alpha-\frac{1}{2})\varepsilon_i^2}\right)$$

$$\leq 4e^{-\varepsilon_i^2/12}.$$

Otherwise, we have

$$(23) = \frac{2(1-\frac{1}{2\alpha})}{\varepsilon_i^2} \int_{\alpha\varepsilon_i^2}^{\infty} \frac{u}{e^{(1-\frac{1}{2\alpha})u}-1}\mathrm{d}u$$

$$\leq \frac{2(1-\frac{1}{2\alpha})}{\varepsilon_i^2} \int_{0}^{\infty} \frac{u}{e^{(1-\frac{1}{2\alpha})u}-1}\mathrm{d}u$$

$$= \frac{2(1-\frac{1}{2\alpha})}{\varepsilon_i^2} \frac{\pi^2}{6(1-\frac{1}{2\alpha})^2}$$

$$\leq \frac{6\beta}{\varepsilon^2}.$$

We now bound $A_2$. As $x \in [-\alpha\varepsilon_i, \alpha\varepsilon_i]$, it comes that $[-(1-\alpha)\varepsilon_i, (1-\alpha)\varepsilon_i] \subset [x-\varepsilon_i, x+\varepsilon_i]$. This implies that

$$A_2 \leq \frac{\int_{-\alpha\varepsilon_i}^{\alpha\varepsilon_i} e^{-x^2/2}\mathrm{d}x}{\int_{-(1-\alpha)\varepsilon_i}^{(1-\alpha)\varepsilon_i} e^{-x^2/2}\mathrm{d}x} - 1$$

$$= \frac{2\int_{(1-\alpha)\varepsilon_i}^{\alpha\varepsilon_i} e^{-x^2/2}\mathrm{d}x}{\int_{-(1-\alpha)\varepsilon_i}^{(1-\alpha)\varepsilon_i} e^{-x^2/2}\mathrm{d}x}$$

$$\leq \frac{2\int_{(1-\alpha)\varepsilon_i}^{\infty} e^{-x^2/2}\mathrm{d}x}{\int_{-(1-\alpha)\varepsilon_i}^{(1-\alpha)\varepsilon_i} e^{-x^2/2}\mathrm{d}x}$$

$$\leq \frac{e^{-(1-\alpha)^2\varepsilon_i^2/2}}{1-e^{-(1-\alpha)^2\varepsilon_i^2/2}} \leq \left(1+\frac{12}{\varepsilon_i^2}\right)e^{-\varepsilon_i^2/12}.$$

The penultimate inequality relies on $\int_x^{\infty} e^{-u^2/2}\mathrm{d}u \leq \sqrt{\frac{\pi}{2}}e^{-x^2/2}$ (see Jacobs and Wozencraft [1965], eq. (2.122)). We obtain again two regimes: $2e^{-\varepsilon_i^2/12}$ if $\varepsilon_i^2 \geq 12$, and $1+\frac{3\beta}{\varepsilon^2}$ otherwise. To summarize, we proved that (21) is bounded by

$$\left(\sqrt{\frac{\pi}{2e}}\frac{2\beta}{\sqrt{\beta-1}}\right)^{|Z'|} \prod_{i\in Z'}\left(\mathbb{I}\left\{\varepsilon^2\frac{4k_i}{\beta} < 12\right\}\left(1+9\frac{\beta}{\varepsilon^2}\right) + \mathbb{I}\left\{\varepsilon^2\frac{4k_i}{\beta} \geq 12\right\}6e^{-\varepsilon^2\frac{k_i}{3\beta}}\right).$$

After the summation on $\mathbf{k}$, on $Z'$, on $q$, and on $Z$, we obtain that there exists two constants $C, C'$ such that

$$\sum_{Z\subset\tau(A^*),\ Z\neq\emptyset} \sum_{q\geq 1} \sum_{Z'\subset Z,\ Z'\neq\emptyset} \sum_{\mathbf{k}\in[q..\infty)^{Z'}} (21) \leq \left(C\varepsilon^{-2}\beta\right)\left(\frac{C'\beta}{\sqrt{\beta-1}}\varepsilon^{-4}\beta^2\right)^{m^*}.$$

$\square$

**Step 2: bound under $\mathfrak{Z}_t \wedge \mathfrak{B}_t$**  The filtered regret bound

$$\mathbb{E}\left[\sum_{t=1}^{T} \Delta_t \mathbb{I}\{\mathfrak{Z}_t \wedge \mathfrak{B}_t\}\right] \leq \frac{n\Delta_{\max}}{p^*}\left(1 + \left(\frac{\Delta_{\min}}{2nB} - \frac{(m^{*2}+1)\varepsilon}{n}\right)^{-2}\right)$$

is obtained as follows. Let $t \geq 1$. First, note that $\mathfrak{B}_t$ implies

$$\left\{\exists i \in \mathrm{T}(A_t) \text{ s.t. } nB_i\big|\overline{\mu}_{i,t-1} - \mu_i^*\big| > \Delta_{\min}/2 - B(m^{*2}+1)\varepsilon\right\}.$$

Then, fixing $i \in [n]$, we can ensure that $i \in S_t$ in the event, using that $p_i(A_t) = \mathbb{P}[i \in S_t | \mathcal{F}_t]$:

$$\mathbb{E}\left[\sum_{t=1}^{T} \mathbb{I}\left\{i \in \mathrm{T}(A_t), \, nB_i\big|\overline{\mu}_{i,t-1} - \mu_i^*\big| > \Delta_{\min}/2 - B(m^{*2}+1)\varepsilon\right\}\right]$$

$$= \mathbb{E}\left[\sum_{t=1}^{T} \frac{p_i(A_t)}{p_i(A_t)}\mathbb{I}\left\{i \in \mathrm{T}(A_t), \, nB_i\big|\overline{\mu}_{i,t-1} - \mu_i^*\big| > \Delta_{\min}/2 - B(m^{*2}+1)\varepsilon\right\}\right]$$

$$= \mathbb{E}\left[\mathbb{E}\left[\sum_{t=1}^{T} \frac{1}{p_i(A_t)}\mathbb{I}\left\{i \in S_t, \, nB_i\big|\overline{\mu}_{i,t-1} - \mu_i^*\big| > \Delta_{\min}/2 - B(m^{*2}+1)\varepsilon\right\}\Big|\mathcal{F}_t\right]\right]$$

$$= \mathbb{E}\left[\sum_{t=1}^{T} \frac{1}{p_i(A_t)}\mathbb{I}\left\{i \in S_t, \, nB_i\big|\overline{\mu}_{i,t-1} - \mu_i^*\big| > \Delta_{\min}/2 - B(m^{*2}+1)\varepsilon\right\}\right]$$

$$\leq \mathbb{E}\left[\sum_{t=1}^{T} \frac{1}{p^*}\mathbb{I}\left\{i \in S_t, \, nB_i\big|\overline{\mu}_{i,t-1} - \mu_i^*\big| > \Delta_{\min}/2 - B(m^{*2}+1)\varepsilon\right\}\right]$$

$$\leq \frac{1}{p^*}\sum_{t\geq 0} 1 \wedge \left(2\exp\left(-2t\left(\frac{\Delta_{\min}}{2nB_i} - \frac{B(m^{*2}+1)\varepsilon}{nB_i}\right)^2\right)\right)$$

$$\leq \frac{1}{p^*}\left(1 + \frac{2\exp\left(-2\left(\frac{\Delta_{\min}}{2nB_i} - \frac{B(m^{*2}+1)\varepsilon}{nB_i}\right)^2\right)}{1 - \exp\left(-2\left(\frac{\Delta_{\min}}{2nB_i} - \frac{B(m^{*2}+1)\varepsilon}{nB_i}\right)^2\right)}\right)$$

$$\leq \frac{1 + \left(\frac{\Delta_{\min}}{2nB_i} - \frac{B(m^{*2}+1)\varepsilon}{nB_i}\right)^{-2}}{p^*}$$

$$\leq \frac{1 + \left(\frac{\Delta_{\min}}{2nB} - \frac{(m^{*2}+1)\varepsilon}{n}\right)^{-2}}{p^*}.$$

**Step 3: bound under $\mathfrak{Z}_t \wedge \mathfrak{D}_t$**  The filtered regret bound

$$\mathbb{E}\left[\sum_{t=1}^{T} \Delta_t \mathbb{I}\{\mathfrak{Z}_t \wedge \mathfrak{D}_t\}\right] \leq \Delta_{\max}\sum_{t\in[T]}\mathbb{E}[\mathbb{P}[\mathfrak{D}_t|\mathcal{H}_t]] \leq \Delta_{\max}\sum_{t\in[T]} 1/T = \Delta_{\max}$$

follows from the following Lemma 3.

**Lemma 3.** *In Algorithm 1 and 2, for all round $t \geq 1$, we have that the probability*

$$\mathbb{P}\left[\big\|\boldsymbol{p}(A_t) \odot \mathbf{B} \odot (\boldsymbol{\theta}_t - \overline{\boldsymbol{\mu}}_{t-1})\big\|_1 \geq \sqrt{2\log(2^n(1 + \lceil\log_2(1/p^*)\rceil)^n T)\sum_{i\in[n]}\frac{p_i(A_t)^2 B_i^2 \beta}{N_{i,t-1}}}\Big|\mathcal{H}_t\right].$$

*is lower than $1/T$.*

*Proof.* We rely on the fact that conditionally on the history, the sample $\boldsymbol{\theta}_t$ is either a Gaussian random vector of mean $\overline{\boldsymbol{\mu}}_{t-1}$ and of diagonal covariance given by $\beta N_{i,t-1}^{-1}/4$ (for Algorithm 2), or a product

of Beta random variable, that is sub-Gaussian with the same covariance matrix [Marchal et al., 2017] (for Algorithm 1). We thus define the functions

$$\alpha_t(A) \triangleq \sqrt{2\log(2^n(1 + \lceil \log_2(1/p^*) \rceil)^n T) \sum_{i \in [n]} \frac{p_i(A_t)^2 B_i^2 \beta}{N_{i,t-1}}},$$

$$\lambda_t(A) \triangleq \frac{2\alpha_t(A)}{\sum_{i \in A} \beta B_i^2 p_i(A)^2 / N_{i,t-1}},$$

we have, with $Q \triangleq \left( \{0\} \cup \{2^{-k}, \ k \in [\lceil \log_2(1/p^*) \rceil] \} \right)^n$,

$$\mathbb{P}\Big[ \big\| \boldsymbol{p}(A_t) \odot \mathbf{B} \odot (\boldsymbol{\theta}_t - \overline{\boldsymbol{\mu}}_{t-1}) \big\|_1 \geq \alpha_t(\boldsymbol{p}(A_t)) \big| \mathcal{H}_t \Big]$$

$$\leq \sum_{\mathbf{q} \in Q} \mathbb{P}\Big[ \mathbf{q} \leq \boldsymbol{p}(A_t) \leq 2\mathbf{q}, \big\| \boldsymbol{p}(A_t) \odot \mathbf{B} \odot (\boldsymbol{\theta}_t - \overline{\boldsymbol{\mu}}_{t-1}) \big\|_1 \geq \alpha_t(\boldsymbol{p}(A_t)) \big| \mathcal{H}_t \Big]$$

$$\leq \sum_{\mathbf{q} \in Q} \mathbb{P}\Big[ \big\| \mathbf{q} \odot \mathbf{B} \odot (\boldsymbol{\theta}_t - \overline{\boldsymbol{\mu}}_{t-1}) \big\|_1 \geq \alpha_t(\mathbf{q})/2 \big| \mathcal{H}_t \Big]$$

$$\leq \sum_{\mathbf{q} \in Q} e^{-\lambda_t(\mathbf{q})\alpha_t(\mathbf{q})/2} \mathbb{E}\Big[ e^{\lambda_t(\mathbf{q}) \| \mathbf{q} \odot \mathbf{B} \odot (\boldsymbol{\theta}_t - \overline{\boldsymbol{\mu}}_{t-1}) \|_1} \big| \mathcal{H}_t \Big]$$

$$\leq \sum_{\mathbf{q} \in Q} e^{-\lambda_t(\mathbf{q})\alpha_t(\mathbf{q})/2} \prod_{i \in [n]} \mathbb{E}\Big[ e^{\lambda_t(\mathbf{q}) q_i B_i |\theta_{i,t} - \overline{\mu}_{i,t-1}|} \big| \mathcal{H}_t \Big]$$

$$\leq \sum_{\mathbf{q} \in Q} e^{-\lambda_t(\mathbf{q})\alpha_t(\mathbf{q})/2} \prod_{i \in [n]} \mathbb{E}\Big[ e^{\lambda_t(\mathbf{q}) q_i B_i (\theta_{i,t} - \overline{\mu}_{i,t-1})} + e^{\lambda_t(\mathbf{q}) q_i B_i (\overline{\mu}_{i,t-1} - \theta_{i,t})} \big| \mathcal{H}_t \Big]$$

$$\leq \sum_{\mathbf{q} \in Q} 2^n e^{-\lambda_t(\mathbf{q})\alpha_t(\mathbf{q})/2} e^{\lambda_t(\mathbf{q})^2 \sum_{i \in A} \beta B_i^2 q_i^2 / (8N_{i,t-1})} \leq 1/T.$$

$\square$

**Step 4: bound under** $\mathfrak{Z}_t \wedge \mathfrak{C}_t \wedge \neg \mathfrak{B}_t \wedge \neg \mathfrak{D}_t$   We get that $\mathbb{E}\Big[ \sum_{t=1}^T \Delta_t \mathbb{I}\{\mathfrak{Z}_t \wedge \mathfrak{C}_t \wedge \neg \mathfrak{B}_t \wedge \neg \mathfrak{D}_t\} \Big]$ is bounded by

$$n\Delta_{\max} + \sum_{i \in [n]} \frac{8(3 + \log(m))\beta B_i^2 \log\Big( 2^n \Big( 1 + \big\lceil \log_2\big(\frac{1}{p^*}\big) \big\rceil \Big)^n T \Big)}{\min_{A \in \mathcal{A}, \ p_i(A) > 0, \ \Delta(A) > 0} \Delta(A)/p_i(A)}$$

from the following derivations. Let $t \geq 1$. Under $\mathfrak{Z}_t \wedge \mathfrak{C}_t \wedge \neg \mathfrak{B}_t \wedge \neg \mathfrak{D}_t$, we have

$$\Delta_t \leq \big\| \boldsymbol{p}(A_t) \odot \mathbf{B} \odot (\boldsymbol{\theta}_t - \boldsymbol{\mu}^*) \big\|_1 + B\Big( m^{*2} + 1 \Big) \varepsilon \qquad\qquad \mathfrak{C}_t$$

$$\leq \big\| \boldsymbol{p}(A_t) \odot \mathbf{B} \odot (\boldsymbol{\theta}_t - \overline{\boldsymbol{\mu}}_{t-1}) \big\|_1 + \big\| \boldsymbol{p}(A_t) \odot \mathbf{B} \odot (\overline{\boldsymbol{\mu}}_{t-1} - \boldsymbol{\mu}^*) \big\|_1 + B\Big( m^{*2} + 1 \Big) \varepsilon$$

$$\leq \big\| \boldsymbol{p}(A_t) \odot \mathbf{B} \odot (\boldsymbol{\theta}_t - \overline{\boldsymbol{\mu}}_{t-1}) \big\|_1 + \Delta_{\min}/2 - B\Big( m^{*2} + 1 \Big) \varepsilon + B\Big( m^{*2} + 1 \Big) \varepsilon \qquad \neg \mathfrak{B}_t$$

$$\leq \big\| \boldsymbol{p}(A_t) \odot \mathbf{B} \odot (\boldsymbol{\theta}_t - \overline{\boldsymbol{\mu}}_{t-1}) \big\|_1 + \Delta_t/2 \qquad\qquad\qquad \mathfrak{Z}_t$$

$$\leq \sqrt{2\log(2^n(1 + \lceil \log_2(1/p^*) \rceil)^n T) \sum_{i \in A_t} \frac{p_i(A_t)^2 B_i^2 \beta}{N_{i,t-1}}} + \Delta_t/2. \qquad \neg \mathfrak{D}_t$$

Thus, the following event holds

$$\mathfrak{A}_t \triangleq \left\{ \Delta_t \leq \sqrt{4\log(2^n(1 + \lceil \log_2(1/p^*) \rceil)^n T) \sum_{i \in A_t} \frac{p_i(A_t)^2 B_i^2 \beta}{N_{i,t-1}}} \right\},$$

and we can apply Lemma 4 to get the bound

$$\mathbb{E}\left[ \sum_{t=1}^T \Delta_t \mathbb{I}\{\mathfrak{A}_t\} \right] \leq n\Delta_{\max} + \sum_{i \in [n]} \frac{(24 + 8\log(m))\beta B_i^2 \log(2^n(1 + \lceil \log_2(1/p^*) \rceil)^n T)}{\min_{A \in \mathcal{A}, \ p_i(A) > 0, \ \Delta(A) > 0} \Delta(A)/p_i(A)}.$$

**Lemma 4** (Adapted from Wang and Chen [2018]). *Let's fix the time horizon $T$. For all $i \in [n]$, let $\beta_{i,T} \in \mathbb{R}_+$. For $t \geq 1$, consider the event*

$$\mathfrak{A}_t \triangleq \left\{ \Delta_t \leq \sqrt{\frac{1}{2} \sum_{i \in [n]} \frac{p_i(A_t)^2 \beta_{i,T}}{N_{i,t-1}}} \right\}.$$

*Let*

$$\delta_{i,\min} \triangleq \min_{A \in \mathcal{A}, \; p_i(A) > 0, \; \Delta(A) > 0} \Delta(A) / p_i(A).$$

*Then, we have*

$$\mathbb{E}\left[ \sum_{t=1}^T \Delta_t \mathbb{I}\{\mathfrak{A}_t\} \right] \leq n\Delta_{\max} + \sum_{i \in [n]} \frac{(3 + \log(m))\beta_{i,T}}{\delta_{i,\min}}.$$

*Proof.* We use the regret allocation method from Wang and Chen [2018]. Specifically, we want to prove that for any time step $t$ where $\mathfrak{A}_t$ holds, we have the following allocation of the regret to each arm $i \in [n]$:

$$\Delta_t \leq \sum_{i \in [n]} g_i(N_{i,t-1}), \tag{24}$$

where the allocation functions are defined for all $i \in [n]$ as

$$g_i(t) \triangleq \mathbb{I}\{t = 0\}\Delta_{\max} + \mathbb{I}\{0 < t \leq L_{i,2}\} \frac{p_i(A_t)\beta_{i,T}^{1/2}}{t^{1/2}} + \mathbb{I}\{L_{i,2} < t \leq L_{i,1}\} \frac{p_i(A_t)\beta_{i,T}}{t\delta_{i,\min}},$$

$$L_{i,1} \triangleq \frac{m\beta_{i,T}}{\delta_{i,\min}^2}, \quad L_{i,2} \triangleq \frac{\beta_{i,T}}{\delta_{i,\min}^2}.$$

Indeed, we can already see that such an allocation produces the bound we are looking for. Notably the following derivation uses, for $i \in [n]$, the equality $p_i(A_t) = \mathbb{E}[\mathbb{I}\{N_{i,t} = N_{i,t-1} + 1\}|\mathcal{F}_t]$, so that for all function $f$, a sum of the form $\sum_{t \geq 1} p_i(A_t)f(N_{i,t-1})$ is equal in expectation to $\sum_{t \geq 0} f(t)$.

$$\mathbb{E}\left[ \sum_{t=1}^T \mathbb{I}\left\{ \Delta_t \leq \sum_{i \in [n]} g_i(N_{i,t-1}) \right\} \Delta_t \right]$$

$$\leq \mathbb{E}\left[ \sum_{t \in [T]} \sum_{i \in [n]} g_i(N_{i,t-1}) \right]$$

$$\leq n\Delta_{\max} + \mathbb{E}\left[ \sum_{i \in [n]} \sum_{t=1}^{L_{i,2}} \frac{\beta_{i,T}^{1/2}}{t^{1/2}} \right] + \mathbb{E}\left[ \sum_{i \in [n]} \sum_{t=L_{i,2}+1}^{L_{i,1}} \frac{\beta_{i,T}}{t\delta_{i,\min}} \right]$$

$$\leq n\Delta_{\max} + \sum_{i \in [n]} 2\sqrt{L_{i,2}\beta_{i,T}} + \sum_{i \in [n]} \left( 1 + \log\left( \frac{L_{i,1}}{L_{i,2}} \right) \right) \frac{\beta_{i,T}}{\delta_{i,\min}}$$

$$= n\Delta_{\max} + \sum_{i \in [n]} \frac{2\beta_{i,T}}{\delta_{i,\min}} + \sum_{i \in [n]} (1 + \log(m)) \frac{\beta_{i,T}}{\delta_{i,\min}}.$$

Therefore, we prove the lemma if we show that $\mathfrak{A}_t$ implies (24). Let $t \geq 1$ be such that $\mathfrak{A}_t$ holds. We can assume that $\Delta_t > 0$ (otherwise, the inequality (24) is trivial). We first observe that

$$\sum_{i \in [n], \; N_{i,t-1} > L_{i,1}} \frac{p_i(A_t)^2 \beta_{i,T}}{\Delta_t N_{i,t-1}} \leq \sum_{i \in [n], \; N_{i,t-1} > L_{i,1}} \frac{p_i(A_t)^2 \beta_{i,T}}{\Delta_t L_{i,1}}$$

$$= \sum_{i \in [n], \; N_{i,t-1} > L_{i,1}} \left( \frac{\delta_{i,\min}}{\Delta_t / p_i(A_t)} \right)^2 \frac{\Delta_t}{m}$$

$$\leq \sum_{i\in[n],\ N_{i,t-1}>L_{i,1}} \frac{\Delta_t}{m}$$

$$\leq \Delta_t. \tag{25}$$

This will be useful to prove that the allocation on all arms $i$ such that $N_{i,t-1} > L_{i,1}$ can be 0. Then we distinguish the following cases:

- If there exists $i_0 \in A_t$ such that $N_{i_0,t-1} = 0$, then

$$\Delta_t \leq \Delta_{\max} = g_{i_0}(N_{i_0,t-1}) \leq \sum_{i\in[n]} g_i(N_{i,t-1}).$$

- If there exists $i_0 \in A_t$ such that $0 < N_{i_0,t-1} \leq p_{i_0}(A_t)^2 \beta_{i_0,T}/\Delta_t^2$, then $N_{i_0,t-1} \leq L_{i_0,2}$ and we have

$$\Delta_t \leq \frac{p_{i_0}(A_t)\beta_{i_0,T}^{1/2}}{N_{i_0,t-1}^{1/2}} = g_{i_0}(N_{i_0,t-1}) \leq \sum_{i\in[n]} g_i(N_{i,t-1}).$$

- If for all $i \in [n]$, $N_{i,t-1} > p_i(A_t)^2 \beta_{i,T}/\Delta_t^2$, then,

$$\sum_{i\in[n],\ L_{i,2}\geq N_{i,t-1}} \frac{p_i(A_t)^2 \beta_{i,T}}{\Delta_t N_{i,t-1}} = \sum_{i\in[n],\ L_{i,2}\geq N_{i,t-1}} g_i(N_{i,t-1}) \frac{p_i(A_t)\beta_{i,T}^{1/2}}{\Delta_t N_{i,t-1}^{1/2}}$$

$$\leq \sum_{i\in[n],\ L_{i,2}\geq N_{i,t-1}} g_i(N_{i,t-1}). \tag{26}$$

On the other hand, using the event $\mathfrak{A}_t$, we have

$$\Delta_t \leq \sum_{i\in[n]} \frac{p_i(A_t)^2 \beta_{i,T}}{2\Delta_t N_{i,t-1}}$$

$$= \sum_{i\in[n],\ L_{i,1}\geq N_{i,t-1}} \frac{p_i(A_t)^2 \beta_{i,T}}{2\Delta_t N_{i,t-1}} + \sum_{i\in[n],\ N_{i,t-1}>L_{i,1}} \frac{p_i(A_t)^2 \beta_{i,T}}{2\Delta_t N_{i,t-1}}.$$

Now, using (25), we get

$$\Delta_t \leq \sum_{i\in[n],\ L_{i,1}\geq N_{i,t-1}} \frac{p_i(A_t)^2 \beta_{i,T}}{2\Delta_t N_{i,t-1}} + \frac{\Delta_t}{2}.$$

We can therefore end the proof in the following way, using (26),

$$\Delta_t \leq \sum_{i\in[n],\ L_{i,1}\geq N_{i,t-1}} \frac{p_i(A_t)^2 \beta_{i,T}}{\Delta_t N_{i,t-1}}$$

$$= \sum_{i\in[n],\ L_{i,2}\geq N_{i,t-1}} \frac{p_i(A_t)^2 \beta_{i,T}}{\Delta_t N_{i,t-1}} + \sum_{i\in[n],\ L_{i,1}\geq N_{i,t-1}>L_{i,2}} \frac{p_i(A_t)^2 \beta_{i,T}}{\Delta_t N_{i,t-1}}$$

$$\leq \sum_{i\in[n],\ L_{i,2}\geq N_{i,t-1}} g_i(N_{i,t-1}) + \sum_{i\in[n],\ L_{i,1}\geq N_{i,t-1}>L_{i,2}} \frac{p_i(A_t)\beta_{i,T}}{\delta_{i,\min} N_{i,t-1}}$$

$$= \sum_{i\in[n],\ L_{i,2}\geq N_{i,t-1}} g_i(N_{i,t-1}) + \sum_{i\in[n],\ L_{i,1}\geq N_{i,t-1}>L_{i,2}} g_i(N_{i,t-1})$$

$$= \sum_{i\in[n]} g_i(N_{i,t-1}).$$

$\square$

# B   Proof of Theorem 2

As for Theorem 1, in the following proof, we will treat both algorithms at the same time. Let's first remark that we can apply steps 1,2 and 3 from Theorem 1, but taking $r_1$ instead of the true reward function. Indeed, letting $E_{j,t} \in \arg\max_{E \in \mathcal{E}_{j,t}} r_j(E, \boldsymbol{\theta}_t)$ and $E_{j,t}^* \in \arg\max_{E \in \mathcal{E}_{j,t}} r_j(E, \boldsymbol{\mu}^*)$ for all $t \geq 1, j \in [\ell]$, we see that the sub-policy playing $E_{1,t}$ at round $t$ is actually minimizing the regret with respect to $r_1$ using CTS with an exact oracle. Since $r_1$ satisfies the assumptions required for Theorem 1, we can apply the steps. Although we still get $\Delta_{\max}$ in these bounds (because the suffered regret remains $\Delta_t$), we notice that $\Delta_{\min}$ is replaced by the minimal gap with respect to $r_1$. To get around this issue, when applying steps 1,2 and 3, we place ourselves under the event that the gap with respect to $r_1$ is greater than $\Delta_{\min}/(2 \sum_{j \in [\ell]} c_j)$. We can thus replace the minimal gap with respect to $r_1$ by this quantity in the bounds. To summarize, we can either place ourselves under the events (for $r_1$) of step 4, obtaining in parallel 3 constant terms, or place ourselves under the event that the gap with respect to $r_1$ is lower than $\Delta_{\min}/(2 \sum_{j \in [\ell]} c_j)$. Our goal now is to do the same for the other reward functions $r_j$. However, since $\mathcal{E}_{j,t}$ can depend on $E_{1,t}, \ldots, E_{j-1,t}$, we define the following filtration

$$(\mathcal{G}_0, \mathcal{G}_1, \ldots, \mathcal{G}_{\ell-1}) = (\mathcal{H}_t, \sigma(\mathcal{H}_t, E_{1,t}), \sigma(\mathcal{H}_t, E_{1,t}, E_{2,t}), \ldots, \sigma(\mathcal{H}_t, E_{1,t}, \ldots, E_{\ell-1,t})).$$

Let us suppose that we have treated the $r_1, \ldots, r_{j-1}$ cases, then, we have at our disposal a filtered approximation regret against events $\mathfrak{Y}_{1,t}, \ldots, \mathfrak{Y}_{j-1,t}$ that $r_1, \ldots, r_{j-1}$ are either in the situation of the step 4 or such the corresponding gap is lower than $\Delta_{\min}/(2 \sum_{j \in [\ell]} c_j)$, respectively. We can write the filtered approximation regret in the following way, by conditioning the expectation with this filtration to get rid of the randomness carried by $\mathcal{E}_{j,t}$:

$$\mathbb{E}\left[ \mathbb{E}\left[ \sum_{t \in [T]} \Delta_t \,\middle|\, \mathcal{G}_{j-1} \right] \mathbb{I}\{\mathfrak{Y}_{1,t}, \ldots, \mathfrak{Y}_{j-1,t}\} \right].$$

We can now apply the same procedure as described above with the reward function $r_j$, on the inner conditional expectation, thus obtaining 3 additional $T$-independent terms and the new filtered approximation regret

$$\mathbb{E}\left[ \mathbb{E}\left[ \sum_{t \in [T]} \Delta_t \mathbb{I}\{\mathfrak{Y}_{j,t}\} \,\middle|\, \mathcal{G}_{j-1} \right] \mathbb{I}\{\mathfrak{Y}_{1,t}, \ldots, \mathfrak{Y}_{j-1,t}\} \right] = \mathbb{E}\left[ \sum_{t \in [T]} \Delta_t \mathbb{I}\{\mathfrak{Y}_{1,t}, \ldots, \mathfrak{Y}_{j,t}\} \right].$$

Therefore, in the end, we have $3j$ constant terms and a filtered approximation regret where all reward functions $r_j$ are in the situation of step 4 or with a corresponding gap lower than $\Delta_{\min}/(2 \sum_{j \in [\ell]} c_j)$. Now, we place ourselves under this event to derive the dominant term of the bound on the approximation regret. We let $J$ be the indices $j$ such that $r_j$ are in the situation of step 4. The derivation of step 4 applied to a function $r_j$ for $j \in J$ gives

$$r_j(E_{j,t}^*, \boldsymbol{\mu}^*) - r_j(E_{j,t}, \boldsymbol{\mu}^*) \leq 2\big\|\boldsymbol{p}(A_t) \odot \mathbf{B}_j \odot (\boldsymbol{\theta}_t - \overline{\boldsymbol{\mu}}_{t-1})\big\|_1.$$

We can therefore use equation (3) to obtain

$$\Delta_t \leq \sum_{j \in [\ell]} \big( r_j(E_{j,t}^*, \boldsymbol{\mu}^*) - r_j(E_{j,t}, \boldsymbol{\mu}^*) \big) \cdot c_j$$

$$\leq 2 \sum_{j \in [\ell]} \left( r_j(E_{j,t}^*, \boldsymbol{\mu}^*) - r_j(E_{j,t}, \boldsymbol{\mu}^*) - \Delta_{\min}/(2 \sum_{j \in [\ell]} c_j) \right) \cdot c_j$$

$$\leq 2 \sum_{j \in J} \left( r_j(E_{j,t}^*, \boldsymbol{\mu}^*) - r_j(E_{j,t}, \boldsymbol{\mu}^*) - \Delta_{\min}/(2 \sum_{j \in [\ell]} c_j) \right) \cdot c_j$$

$$\leq 4 \sum_{j \in J} c_j \big\|\boldsymbol{p}(A_t) \odot \mathbf{B}_j \odot (\boldsymbol{\theta}_t - \overline{\boldsymbol{\mu}}_{t-1})\big\|_1.$$

From there, we can repeat the end of the proof of Theorem 1 with the weight in front of each arm $i$ being $\sum_{j \in [\ell]} B_{ij} c_j$.

## C Travelling salesman problem (TSP)

Here we give another example of a problem that falls into the REDUCE2EXACT setting, namely the *Travelling salesman problem* (TSP). We will see that unlike the examples mentioned before, here we need to modify the algorithm a bit to fully fall within the REDUCE2EXACT setting.

Given a complete undirected weighted graph $G = (V, E)$ whose distances $\boldsymbol{\mu}^* \triangleq (d(u,v))_{(u,v) \in E}$ have to satisfy the triangle inequality, the goal is to find an Hamiltonian cycle $A \in \mathcal{A} \triangleq \left\{ \{(v_0, v_1), \ldots, (v_{|V|-1}, v_{|V|})\} : \{v_1, \ldots, v_{|V|} = v_0\} = V \right\}$ of minimum cost $\sum_{i \in [|V|]} d(v_{i-1}, v_i)$. We consider the following oracle from the Christofides [1976] algorithm ($\alpha = 2/3$).

- $\text{Oracle}_1(\boldsymbol{\mu})$ : The algorithm of Christofides combines $\mathcal{E}_1 \triangleq \{\text{spanning trees of } G\}$ and $\mathcal{E}_2 \triangleq \{\text{perfect matchings of the subgraph } G' \text{ of } G \text{ induced by the vertices of odd order in } E_1\}$, with $r_1$ being the weight of the spanning tree and $r_2$ the weight of the perfect matching.
- $\text{Oracle}_2(E_1, E_2)$ : $\text{Oracle}_2$ combines the edges of $E_1$ and $E_2$ to form a connected multigraph $\widetilde{G} = \left( \widetilde{V}, \widetilde{E} \right)$ in which all vertices have even degree (so it is Eulerian), forms an Eulerian circuit in this multigraph, and finally, outputs the Hamiltonian cycle obtained by skipping repeated vertices (shortcutting).

Thanks to the triangle inequality, shortcutting does not increase the weight, so we have

$$\mathbf{e}_A^\intercal \boldsymbol{\mu}^* \leq r_1(E_1, \boldsymbol{\mu}^*) + r_2(E_2, \boldsymbol{\mu}^*).$$

Let's now deal with $A^*$. Removing an edge from $A^*$ produces a spanning tree, so $\mathbf{e}_{A^*}^\intercal \boldsymbol{\mu}^* \geq r_1(E_1^*, \boldsymbol{\mu}^*)$. On the other hand, by the triangle inequality, the weight of the optimal TSP solution for $G'$ is lower than $\mathbf{e}_{A^*}^\intercal \boldsymbol{\mu}^*$ (visiting more nodes does not, in any case, reduce the total cost). Taking every second edge of this cycle (which is of even length since all graphs have an even number of vertices of odd degree) we obtain a matching that has a weight less than half the weight of the cycle (if this is not the case we can take the complementary), so $\mathbf{e}_{A^*}^\intercal \boldsymbol{\mu}^*/2 \geq r_2(E_2^*, \boldsymbol{\mu}^*)$. To summarize, we have

$$\frac{2}{3} \mathbf{e}_A^\intercal \boldsymbol{\mu}^* - \mathbf{e}_{A^*}^\intercal \boldsymbol{\mu}^* \leq \frac{2}{3}(r_1(E_1, \boldsymbol{\mu}^*) - r_1(E_1^*, \boldsymbol{\mu}^*) + r_2(E_2, \boldsymbol{\mu}^*) - r_2(E_2^*, \boldsymbol{\mu}^*)).$$

From the above, we can see that all the criteria of REDUCE2EXACT are satisfied, except Assumption 2. Indeed, we assume that we receive feedback from $A$, while we would need feedback from the set $\widetilde{E}$. We could probably have foreseen that the TSP would pose a difficulty in our assumption: indeed, among the many operations performed by the oracle to build the final solution, shortcutting is the one that does not imply an optimization, but rather makes the solution feasible (so it does not represent a sub-problem as we have defined it in this paper). In other words, if we allowed the tour to pass over the same vertex several times, then the TSP would belong to REDUCE2EXACT by skipping the shortcutting step. Yet, we note that a workaround is possible by by taking a closer look at the shortcutting step: in this step, we have an Eulerian circuit that we follow by skipping some edges. Even if the skipped edges are replaced by new ones such that the distance traveled decreases, it is precisely the absence of feedback on the skipped edges that poses an issue. Therefore, for a given edge of the Eulerian circuit, it would be helpful to be able to guarantee that some feedback is obtained on this edge, i.e., that it has a chance to belong to the final Hamiltonian cycle. The trick is then to notice that the shortcutting step depends on the edge from which we start. In particular, this edge is guaranteed to be in the final Hamiltonian cycle. This choice of the first edge is generally presupposed to be arbitrary and has no influence on the guarantees obtained previously, but for us, it can be used to force an edge to produce a feedback. More precisely, to choose this first edge, we can use a uniform randomization on $\widetilde{E}$, meaning that for each edge $e \in \widetilde{E}$, the probability $q_e$ that $e$ belongs to $A$ is such that $q_e \geq \frac{1}{|\widetilde{E}|}$. Notice that since $\widetilde{E}$ is a multigraph edge set, for a "true" edge $e \in E$, we can have two edges $e_1, e_2 \in \widetilde{E}$ representing it and in that case $q_{e_1} + q_{e_2} = p_e(A)$. We thus get our Assumption 4:

$$|r_1(E_1, \boldsymbol{\mu}) + r_2(E_2, \boldsymbol{\mu}) - (r_1(E_1, \boldsymbol{\mu}') + r_2(E_2, \boldsymbol{\mu}'))| \leq \sum_{e \in \widetilde{E}} |\mu_e - \mu'_e| \leq \left| \widetilde{E} \right| \sum_{e \in \widetilde{E}} q_e |\mu_e - \mu'_e|$$

$$= \left| \widetilde{E} \right| \sum_{e \in E} p_e(A) |\mu_e - \mu'_e|.$$