# OpenReview forum: "When Combinatorial Thompson Sampling meets Approximation Regret"
_NeurIPS.cc/2022/Conference — NeurIPS 2022 Accept_

### Official Review · Reviewer_4G2X · 2022-07-11

**Rating:** 6
**Confidence:** 1
**Soundness:** 3 good
**Presentation:** 2 fair
**Contribution:** 3 good

**Summary:**

The authors study stochastic combinatorial Thompson Sampling (CTS), with semi-bandit feedback in a setting where exactly solving the corresponding offline optimization problem is not feasible. While existing work provided approximation regret analysis of CTS for the greedy oracle, the proposed work provides a regret upper bound for CTS, obtained under a specific condition on the approximation oracle. The authors present various examples where the specific conditions are satisfied and counter-examples.

The paper identifies assumptions (mainly 2 and 4) under which the problem becomes exact with O(log T/\Delta).  Contribution is more on identifying these assumptions than the associated regret analysis.

**Questions:**

Is Assumption 2 verifiable?

**Limitations:**

yes

**Strengths And Weaknesses:**

Strength:

The paper identifies assumptions (mainly 2 and 4) under which the problem becomes exact with O(log T/\Delta).

Weakness:

Since the condition reduces to the exact oracle, I am not sure whether it is correct to say that it is the first  O(log T/\Delta) approximation regret.

Writing style can be improved.

---

> ### Author Response · Authors · 2022-07-31
> **We respond to your review regarding the correctness of our contribution statement and the verifiability of Assumption 2.**
>
> Thank you for your review. We actually think it is correct to say that this is the first $\mathcal{O}(\log T/\Delta)$ approximation regret bound because even if the condition enables us to reduce to the exact oracle in the analysis, the final result is definitely on the approximation regret. Prior to our work, no $\mathcal{O}(\log T/\Delta)$ approximation regret bound existed.
>
> Assumption 2 is generally provable without much difficulty. Since $r$ is the expectation of a function of $S$, it is sufficient to prove a Lipschitz condition with $\left\|\left\|~\right\|\right\|_1$ on this function to obtain Assumption 2.

---

### Official Review · Reviewer_6J2J · 2022-07-11

**Rating:** 7
**Confidence:** 5
**Soundness:** 3 good
**Presentation:** 2 fair
**Contribution:** 3 good

**Summary:**

This work studies the behavior of the Combinatorial Thompson Sampling policy (CTS) for the probabilistically triggered arms extension of CMAB (CMAB-T). They propose a specific condition for the approximation oracle: REDUCE2EXACT based on the intuition of reduction, and show some problems using the approximation oracle such as the greedy oracle and the LP relaxation oracle which meet the REDUCE2EXACT assumption. This novel decomposition of the approximation oracle is helpful for the analysis of the approximation oracle by CTS. For example, when the reward function is monotone submodular and the oracle is the greedy oracle, it can avoid the "mismatch" phenomenon mentioned in the previous work and obtain a better regret upper bound. Under the REDUCE2EXACT assumption, this paper proves the first $O(\text{log}(T)/\Delta)$ approximate regret upper bound for CTS. In addition, this work also provides a counter-example for a deeper understanding of REDUCE2EXACT.

**Questions:**

Please also see the weakness part.

It seems that when the approximation oracle is greedy, the reward function must be monotone submodular. Can you give an example of the reward function without too many assumptions and meeting the REDUCE2EXACT assumption?

Suggestion:

1. The process will be clearer if useful notations are defined.

2. It will be better if this work can describe the process of OIM meeting the REDUCE2EXACT assumption in detail. This will help readers understand how REDUCE2EXACT works under CMAB-T.


**Strengths And Weaknesses:**

Strengths:
1. This work is novel in the understanding and analysis of the approximation oracle, and is helpful for the analysis of CTS of the approximation oracle. Under REDUCE2EXACT assumption, this work proves a better regret upper bound compared with Kong et al. [2021].

2. This work provides a counter-example for the REDUCE2EXACT assumption, which clearly describes the limitations of REDUCE2EXACT assumption.

Weakness：

1. I did not see too many difficulties in the regret analysis process of Theorem 1. It seems that the proof process of Theorem 1 is a reorganization of the proof processes of Huyuk and Tekin [2019], Perrault et al. [2020a] and Perrault [2020]. It is worth noting that this work does not describe the difficulties or non-trivial tricks of the proof after changing the CMAB setting to CMAB-T.

2. This work is a follow-up work based on Kong et al. [2021], but in this work it is not described how they avoid the "mismatch" phenomenon mentioned in Kong et al. [2021]. This may confuse readers.

3. The abuse of notation: In the appendix, there is the abuse of notation due to the need to cite other works. Specifically, there are some notations defined in other articles that are not defined in this work, such as $\tau_j$, $Z^c$; and there are also some notations that I have not seen defined in the works they refer to, such as $ \eta_{q,k}$.

---

> ### Author Response · Authors · 2022-07-30
> **Your comments are very useful to us. We provide an overview of your main point about an application of REDUCE2EXACT to a more general example, considering the case of a submodular non-monotonic reward function.**
>
> Thank you for your review. Your remarks are relevant and appreciated.
>
> We will describe the non-trivial trick of the proof of Theorem 1 after changing the CMAB setting to CMAB-T, referecing notably to line 500.
>
> Concerning the mismatch phenomenon, this point is indeed very relevant. The phenomenon is avoided in the virtue of Assumption 4, which links the gap to be distinguished and the actually paid gap. We will add this crucial point to the paper.
>
> We will definitely follow your suggestions, namely the definition of useful notations and the detail of the process of OIM meeting the REDUCE2EXACT assumption.
>
> About giving an example of a reward function without too many assumptions and meeting REDUCE2EXACT, it is not hard to imagine a submodular reward function that does not respect the monotonicity assumption, while verifying the REDUCE2EXACT assumption. Actually, Max-Cut is such an example of non-monotone submodular maximization problem. To answer your question more generaly, we can consider the general maximization of a non-monotonic submodular function. The $2/5$ approximation algorithm of https://people.csail.mit.edu/mirrokni/focs07.pdf is an example that can be considered. We see no conceptual reason that would prevent such an oracle from verifying REDUCE2EXACT.

---

> > ### Comment · Reviewer_6J2J · 2022-08-07
> > **Thank you for the reply**
> >
> > The example you added with a submodular non-monotonic reward function is correct and makes sense. I understand the scope of the REDUCE2EXACT assumption better with this example.
> >
> > As stated in the review & rebuttal, the writing of this paper needs to be polished. I have the following suggestions.
> >
> > (1) In the beginning, the Greedy Oracle can be used to describe the intuition of the REDUCE2EXACT assumption and explain how to avoid the "mismatch" phenomenon. Next is to extend the assumptions to other forms of Oracle and reward functions.
> >
> > (2) For Theorem 1, while describing the non-trivial trick, it is also necessary to highlight the limitation of Theorem 1, such as the existence of $1/p^*$ and the difficulties in transitioning from the gap-dependent upper bound to a gap-independent upper bound.

---

> > > ### Author Response · Authors · 2022-08-07
> > > **Your suggestions are insightful**
> > >
> > > Thank you for your suggestions.
> > >
> > > (1) Yes, it is clear that the Greedy Oracle is a good example to give the insights of the assumptions. When you say at the beginning, do you mean at the beginning of section 4, or at the beginning of the article in the "Contributions" paragraph?
> > >
> > > (2) Talking about limitations and challenges right after the statement of Theorem 1 is indeed a good idea.

---

### Official Review · Reviewer_aZHQ · 2022-07-11

**Rating:** 7
**Confidence:** 4
**Soundness:** 3 good
**Presentation:** 2 fair
**Contribution:** 3 good

**Summary:**

This paper studies the combinatorial Thompson sampling (CTS) algorithm for CMAB-T with approximate oracles. They first show that CTS can achieve O(log T/\Delta) regret for CMAB-T problem with exact oracles. As for the general approximate oracle, they introduce an assumption and show that once the assumption holds, the alpha-approximate regret can be decomposed by the sum of several exact regrets. And thus the upper bound for the \alpha-approximate regret can be derived. This is also the first guarantee for alpha-approximate regret of the CTS algorithm.

**Questions:**

Please see above weakness.

**Limitations:**

The work is about online learning theory and does not have negative societal impact.

**Strengths And Weaknesses:**

Strength:
This paper makes progress on the open question that whether the CTS algorithm can achieve sub-linear alpha-approximate regret guarantee. By establishing an assumption which is satisfied in common applications, they show that the alpha-approximate regret can be decomposed as the sum of several exact regrets. And these exact regrets can be simply upper bounded.
They find several examples such as greedy and LP that satisfy the proposed assumption.
Weakness:
In fact, the constant term in current regret upper bound has an additional 1/p* term, where p* can be exponentially small since the triggering probability can be small. It seems that such a term appears mainly due to the Lipschitz condition without using the TPM condition. So I want to ask that would it possible to avoid the 1/p* term in the regret?
The authors should also compare the derived regret upper bound with the upper bound for CUCB.
The presentation is not easy to follow, especially for the description of Assumption 4 and the analysis of PMC. I think more explanations are required for better understanding.

Typo:
Line 192: depends -> depend

---

> ### Author Response · Authors · 2022-07-30
> **The main point of your review identifies an open question about $1/p^*$.**
>
> Thank you for the points you raised in your review. Concerning the $T$-independent term, you are right, the additive constant has a $1/p^*$ dependence, and you have indeed identified the cause: the use of the Lipschitz condition without weighting by probabilities. We think indeed that this dependence should be avoidable, but it seems that another technique should be considered. From our knowledge, this is still an open question. By the way, in http://proceedings.mlr.press/v89/huyuk19a/huyuk19a.pdf, this constant term is also present (but is unavoidable as they do not have the probabilities in their Lipschitz condition). We will add this remark in the limitations. Maybe the triggering probability groups from https://arxiv.org/pdf/1703.01610.pdf can help to get around this issue, and it is probably worth looking into this further.
>
> We will add a comparison with the regret upper bound for CUCB (https://arxiv.org/pdf/1703.01610.pdf), where the $1/p^*$ dependence is avoided, but where the structure of the dependences between the arms cannot be exploited to have a tigther leading term.

---

### Official Review · Reviewer_DcCA · 2022-07-11

**Rating:** 4
**Confidence:** 3
**Soundness:** 3 good
**Presentation:** 1 poor
**Contribution:** 3 good

**Summary:**

This paper concerns the performance of Thompson sampling for combinatorial semibandits with an imperfect \alpha-approximate oracle for the underlying combinatorial optimization problem. It shows regret bounds compared to an alpha-approximation to optimality, under a structural assumption on this oracle which is satisfied for e.g. greedy algorithms for submodular objectives.

**Questions:**

Major comments:


1. I found the setup regarding \mu very confusing. At the start of page 2, when \mu is used for the first time, it would be helpful to explain that it is just a model parameter. Later at the top of page 4, it was unclear to me why \mu should be n-dimensional for n the number of arms. In fact if I understand correctly, this is not the case for the first PMC example later in the paper (I believe there, \mu is indexed by edges of a graph while arms are indexed by vertices). This might affect Assumption 2 which as stated requires \mu to be n-dimensional.


2. Why is 1/\Delta_{i,min} a reasonable parameter for regret to depend on? I would typically expect it to be exponentially large in n, e.g. for a random choice of \alpha. If it is possible to have a sqrt(T) type regret bound I would find it much more convincing.



Minor comments:


1. In the Contributions paragraph, it could be made clearer that the condition on Oracle is structural and applies to greedy algorithms for submodular optimization. Otherwise, it sounds like maybe this condition is "exogenous" and only holds in extremely specific cases, or that the assumption amounts to bounding some problem-dependent parameter. Basically, I found the result more interesting than I expected to from the introduction!

Also in this paragraph, I would reference Theorem 2 and the following examples when appropriate so the claims are not just flying around in this paragraph.

2. I believe this paper is targetting frequentist regret for TS, but it could be made clearer since often TS is analyzed with Bayesian regret.


3. The setup at the start of section 2 is confusing in several ways. "A_t\in\mathcal F_t" does not make sense to me, I imagine the intended meaning is that A_t is \mathcal F_t measurable.


4. Line 141: "assumptions are often made struck me as an unconvincing justification.

5. Line 146: please define p_i outside of this assumption.

6. Definition 2: please use way more lines here. It is pretty miserable to read this as currently formatted.


In fact, in the later examples, more space around the definitions of Oracle_1 and Oracle_2 would be highly appreciated as well.


7. "Exact subproblem" was confusing to me. I think what is intended is that some sub-optimization-problem is solved exactly. I would just write this out in more words. Similarly, I found "exact sub-regrets" a confusing terminology. In some areas of math, "exact" is a serious technical term!

Also the first paragraph of Section 4 didn't make sense to me. Perhaps it could be clarified, I'm not sure.


8. It seems to be that "Assumption 4" should usually say "Assumption 4 with parameter vector c". It is confusing as written because c does not appear in Assumption 4 until Oracle_2, but c does appear in e.g. the statement of Theorem 2. I would introduce c at the start of Assumption 4.


Actually in general, Assumption 4 felt frustrating to read. For example, when I read the definition of Oracle_1, I first see "There are \ell reward functions (r_j)_{j\in[\ell]}." At this point, I have no idea what spaces r_j are defined on, what their codomains are, etc. Mathematical definitions should never work this way. (Also at the end of the definition, it is still unclear what the codomain of r_j is...). I would say something more like:

"Oracle_1(\mu) must output a sequence (E_1,...,E_{\ell]) described as follows. For each j\in [\ell], let \mathcal E_j=\mathcal E_j(E_1,\dots,E_{j-1}) be a finite subaction space which may depend on E_1,...,E_{j-1}. Then we require that E_j\in \arg\max_{E\in\mathcal E_j} r_j(E,\mu).

Informally, Oracle_1(\mu) exactly solves a finite sequence of recursively defined optimization subproblems. Note that the subproblems may have multiple optimal solutions, so Oracle_1(\mu) may output many values for each given \mu."


9. In line 226, it is unclear where the definition of Oracle_2 ends.

10. Many minor grammatical issues especially around singular/plural. I am not taking this much into account for my score since the authors might not be native English speakers, but it would improve readability to fix them. E.g.:

	- Line 134: "outcomes" -> "outcome"

	- Line 135-136: A, D_{trig}, r don't seem to be "quantities", which should refer to actual numbers.

	- Line 202: "Assumptions" -> "Assumption", "have" -> "has".


11. For many of the references, "thompson sampling" is in lower case when it should not be. Use double brackets {{ }} around the title in the bibtex to fix this.


**Limitations:**

--

**Strengths And Weaknesses:**

The technical content of this paper looks nice and possibly accept worthy. The authors show that for combinatorial optimization problems with natural structure, Thompson sampling does achieve good approximation regret, even though counterexamples without this structure do exist. I found the result illuminating and creative (though I haven't thought about this issue in the past). I do have some qualms about the notion of gap that is used for regret as it seems potentially exponentially small (mentioned below in the questions section).

However I think the writing is really quite confusing in many places, which is the main reason for my low score. I understand that the page limit is rather stringent, so my high-level suggestion for the authors in the future would be to move some later content (e.g. examples and the TSP counterexample) to the appendix to make space for proper exposition.

---

> ### Author Response · Authors · 2022-07-29
> **Your review will help us in the re-writing. We respond to your main comments.**
>
> Thank you for your valuable review. We will take all your points into account in the revised version of the paper. We do realize that the writing is very confusing in several places. We will clearly use your feedback to remedy this.
>
> Here are the answers to the main comments/questions.
>
> The vector ${\boldsymbol \mu}^*$ is actually introduced earlier via $\mu_i^*$. We will explicitely state at the start of page 2 that it is the model parameter. It is indeed always $n$-dimensional, where $n$ is the number of base arms. For us, an arm is only something that produces an outcome - not something you can choose as an action (this is the role of the action space: an action triggers some arms). Thus, for the PMC problem, there are $n=|E|$ arms. We will clarify this point.
>
> $\Delta_{i,\min}$ is a usual quantity for the regret bound. It can be arbitrarily small for a certain choice of $\alpha$, just like the standard exact gap can be arbitrarily small for a certain ${\boldsymbol \mu}^*$. This kind of bound can be transformed into a $\sqrt{T}$ bound, see e.g., Theorem 2 from http://proceedings.mlr.press/v28/chen13a.pdf. For CTS, achieving this transformation is still an open question. We will add this point to the paper.
>
> We will state the following in the Contributions paragraph:
>  - The condition on Oracle is structural and applies to greedy algorithms for submodular optimization.
>  - We target frequentist regret.
>  - Our main result is Theorem 2, as well as its application in several examples like Probabilistic maximum coverage and Online influence maximization.
>
> We will revise the terminology Exact subproblem/subregret, and rework the writing of Assumption 4 to make it more formal, following your proposition. We will also detail the paragraph before Assumption 4.

---

### Meta-Review · Area_Chair_iGDN · 2022-08-22

**Recommendation:** Accept
**Confidence:** Less certain

**Metareview:**

We thank the authors for their submission.

This work considers combinatorial multi-armed bandits, in which the agent chooses a subset of arms at each round and receives some combinatorial function of the mean rewards of her chosen arms.
Specifically, it studies Thompson sampling algorithms in which the only access to the underlying combinatorial problem is via an offline approximation oracle. The authors show novel no-approximate-regret algorithms whose regret guarantees hold under mild assumptions on the oracle (applicable for many common combinatorial problems).

The paper, however, has some apparent writing issues which we ask the authors to address in its camera-ready version.

**Award:**

No

---

### Decision · Program_Chairs · 2022-09-14

Accept